# Probabilistic causal reasoning under time pressure

Ivar R. Kolvoort[1,2,3]*, Elizabeth L. Fisher[1,4], Robert van Rooij[2], Katrin Schulz[2], Leendert van Maanen[3]

1 Department of Psychology, University of Amsterdam, Amsterdam, The Netherlands, 2 Institute for Logic, Language, and Computation, University of Amsterdam, Amsterdam, The Netherlands, 3 Department of Experimental Psychology, Utrecht University, Utrecht, The Netherlands, 4 Turner Institute for Brain and Mental Health, School of Psychological Sciences, and Cognition & Philosophy Laboratory, Monash University, Clayton, Australia

* ivarrenzokolvoort@gmail.com

## Abstract

While causal reasoning is a core facet of our cognitive abilities, its time-course has not received proper attention. As the duration of reasoning might prove crucial in understanding the underlying cognitive processes, we asked participants in two experiments to make probabilistic causal inferences while manipulating time pressure. We found that participants are less accurate under time pressure, a speed-accuracy-tradeoff, and that they respond more conservatively. Surprisingly, two other persistent reasoning errors—Markov violations and failures to explain away—appeared insensitive to time pressure. These observations seem related to confidence: Conservative inferences were associated with low confidence, whereas Markov violations and failures to explain were not. These findings challenge existing theories that predict an association between time pressure and all causal reasoning errors including conservatism. Our findings suggest that these errors should not be attributed to a single cognitive mechanism and emphasize that causal judgements are the result of multiple processes.

**Data Availability Statement:** Data and analysis code for the experiments in this paper has been made publicly available and are hosted by the Open Science Framework at https://osf.io/bz9vj/.

## 1. Introduction

Humans are expert causal reasoners, even though they might not be explicitly aware of it. The point that causal judgements play a role in many decisions, has been made many times before [e.g. 1, 2]. Nevertheless, it is a point worthy of reiterating here: Most actions are based on perception of and reasoning about causes and effects in the world. How will your colleagues react if you are late for that meeting? What are the chances of getting the flu knowing your flatmate has it? The answers to these and most similar questions depend on your beliefs about how events are causally related.

Numerous experiments have shown that causal reasoning affects categorization, category-based inferences, learning, prediction, as well as decision making [2–5]. Beliefs about causal structures are crucial in decision making. Someone who planned to go mountaineering next week might decide to stay at a hotel to decrease the risk of catching the flu from a flatmate.

**Funding:** This study was supported by an Interdisciplinary Doctoral Agreement grant by the University of Amsterdam awarded to KS, RvR, and LvM. The funder had no role in study design, data collection and analysis, decision to publish, or preparation of the manuscript.

**Competing interests:** The authors have declared that no competing interests exist.

However, had she believed that catching the flu is not caused by exposure to the flu virus, then she would have decided to stay at home. This is just one example of how beliefs about causal relationships impact decision-making. Causality ties into most of what we do and think, which is why the topic of causality has received more and more attention from cognitive scientists in the last decades.

Causal Bayesian Networks (CBN; also known as Causal Graphical Models) have become the dominant framework for modelling probabilistic causal phenomena. CBNs have been used in many scientific disciplines as a normative framework to make predictions about causal phenomena [6–8]. Besides being used as a normative framework, CBNs have also been used as psychological models of causal reasoning [9–18], of causal learning [19–26], and of categorization [27–31]. CBN is a normative theory in that, under the assumption that the structure and parametrization of a graph correspond truthfully to the world, the inferences the model allows for correspond truthfully to the world. CBN provides a reasonable description of human causal judgements [2, 11]. CBN accurately predicts that people are susceptible to subtle changes in graph structure and parametrization [2, 9, 10, 18, 32, 33]. However, human causal judgments do not appear to be fully in line with the normative model. Instead, they deviate persistently and systematically from the normative CBN prediction.

## 1.1 Reasoning errors

There are three specific reasoning errors (i.e. deviations from CBN predictions) people are known to commit: violations of Markov independence, failures to explain away, and conservative inferences [2, 33]. These deviations from the CBN model are important as they can provide insight in the cognitive processes involved in causal judgments. In the remainder of this section and the rest of this manuscript we will restrict our focus on binary causal variables with generative causal relationships. Such a setting has been used as the standard in the literature on causal judgments and simplifies our discussion.

People have been found to systematically violate the principle of Markov independence ('Markov violations') in a variety of experimental paradigms and regardless of how they learn about a causal network [2, 4, 9, 18, 32–38]. Markov independence refers to the independence of certain events within a causal structure. For instance, with a common cause structure $X_1 \leftarrow Y \rightarrow X_2$ (Fig 1A) people often think that the state of $X_1$ is relevant in any situation when inferring $X_2$. However, $X_1$ is only relevant here when Y is unknown. When Y is known, information about $X_1$ does not provide additional information about $X_2$, as Y completely mediates the effect of $X_1$ on $X_2$. The exact same holds for a chain structure (Fig 1B).

In these cases we can state Markov independence formally as
$P(X_i = x | Y = y, X_j = 0) = P(X_i = x | Y = y, X_j = 1) = P(X_i = x | Y = y)$, where the subscripts i and j refer to the two X variables, and the values 0 and 1 refer to a variable being absent or

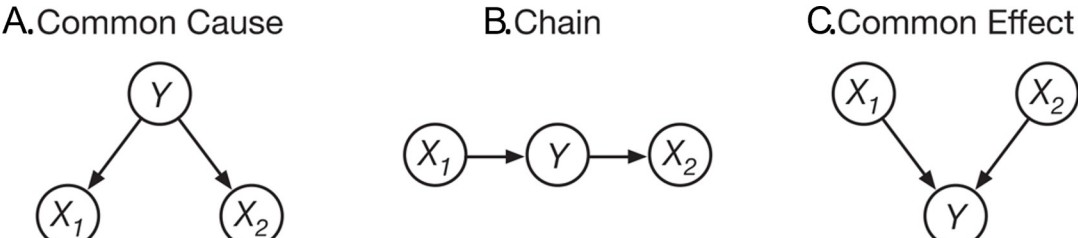

**Fig 1. Three-variable causal network structures.** The circles represent causal variables and the arrows the causal relationships between them.

present respectively (we will use this notation throughout this paper). Markov independence also holds in a *common effect* structure (Fig 1C), where the causes are independent when the effect is not known, such that $P(X_i = x|X_j = 1) = P(X_i = x|X_j = 0) = P(X_i = x)$. Instead of adhering to the principle of Markov independence, people tend to judge that $P(X_i = 1|Y = y, X_j = 1) > P(X_i = 1|Y = y) > P(X_i = 1|Y = y, X_j = 0)$ in common cause and chain structures, and $P(X_i = 1|X_j = 1) > P(X_i = 1) > P(X_i = 1|X_j = 0)$ in the common effect structure [2, 4, 9, 18, 32, 33, 36–38]. In common cause and chain structures this is also known as 'failures in screening-off', which refers to the fact that the dependence between two variables is 'screened-off' by a third variable, namely Y in our notation. A second reasoning error is related to the principle of *explaining away* (also known as 'discounting'). Explaining away is involved in situations where multiple causes can independently bring about an effect and a judgment is required about the actual cause of the effect. Imagine a situation in which a friend has a headache and you know that the (only) two possible causes for this are alcohol consumption and the flu (a common effect structure, Fig 1C). Now, upon learning that your friend consumed alcohol last night it becomes less likely that they have the flu. This is because alcohol consumption 'explains away' the presence of a headache. Reversely, if you learn that your friend did not consume alcohol, it makes it more likely that they have the flu, as some other cause than alcohol consumption must have brought about the headache.

Put more generally, explaining away refers to cases in which a target cause (flu in the previous example) becomes less (more) likely after learning about the presence (absence) of another cause (alcohol consumption in the previous example, we refer to this as the non-queried cause). Referring to the common effect structure in Fig 1C, we can state explaining away formally as $P(X_i = 1|Y = 1, X_j = 0) > P(X_i = 1|Y = 1) > P(X_i = 1|Y = 1, X_j = 1)$. Multiple studies have found that people engage in insufficient explaining away compared to the CBN prediction or that they do not explain away at all [2, 33, 38–42].

The third persistent reasoning error, *conservatism*, refers to a tendency of people to not give 'extreme' responses, but rather to respond somewhere near the middle of a response scale. For probability judgements this means that extreme responses near 0% or 100% are often avoided and that people prefer to respond closer to 50%. In their review of experiments on inferences from causal networks, Rottman and Hastie [2] concluded that many inferences were conservative relative to the CBN prediction [see 41, 43–45]. They found that responses are generally between 50% and the CBN prediction, which indicates that participants are not sensitive enough to the parameters of causal networks. In later work Rottman and Hastie [33] found that judgements were particularly close to 50% when the state of one variable in the network was unknown ('ambiguity trials', such as $P(X_i = 1|Y = 1)$) or when the two other variables provided conflicted cues ('conflict trials', such as $P(Y = 1|X_i = 1, X_j = 0)$).

It is important to investigate in what situations these reasoning errors are prominent and how they come about as this can shed light on the processes underlying causal reasoning. One fruitful way to investigate these errors and accuracy in causal judgements more generally is by utilizing time pressure.

## 1.2 Time pressure

It stands to reason that when participants have less time available to provide causal judgments, behavior will deteriorate in specific ways. In other domains within the larger field of judgement and decision making, the analysis of response time (RT) and the explicit use of time pressure manipulations has led to a better understanding of the cognitive processes involved. Examples include perceptual decision-making [46–48], economic decision-making [49–52],

judgement under uncertainty [53–56], probabilistic reasoning [57, 58], and syllogistic reasoning [59, 60].

In contrast, within the causal reasoning literature RT, measurements and time pressure manipulations have received little attention. While the effect of time on causal structure learning has been studied a few times [e.g. 21, 61], we are aware of only one study involving time pressure that directly pertains to causal probabilistic inferences. Experiment 2 in [18] asked participants to judge under time pressure in which causal network a certain variable value was more likely, but no relevant effects of RT or time pressure were found. Importantly though, it was pointed out that this study could only be considered preliminary [62]; the time pressure manipulation was possibly ineffective and the specific task they used was complex and did not generalize to other paradigms as it required the comparison of two causal configurations.

As time pressure manipulations and RT analysis have enabled significant development in other domains of cognitive science, we here aim to use these methods to spur similar development in our understanding of causal reasoning. Before getting to the application of these methods, let us first discuss some important aspects of them.

**1.2.1 Speed-accuracy tradeoff (SAT).** A crucial phenomenon used in the study of time pressure is the speed-accuracy tradeoff (SAT) [63–66]. The SAT refers to the common observation that accuracy tends to covary with response times [64]. The typical observation is that faster responses are less accurate. This pattern has been observed across individuals [67–69], across conditions (macro-SAT) [70, 71], and within conditions (micro-SAT) [72, 73].

While the SAT is often used to refer to the overall accuracy of responses, we can apply the same idea to different ways of measuring accuracy. In the case of causal judgments, we can apply notions of micro- and macro-SAT to the three reasoning errors discussed previously. That is, we can investigate whether the magnitude of these errors changes under an external time pressure manipulation (macro-SAT), and whether they are associated with RTs within conditions (micro-SAT), i.e. the passing of time or 'internal' time pressure.

The effects of time pressure, including micro- and macro-SATs, on causal reasoning have not been explicitly studied. Therefore, the main aim of Experiment 1 was to explore how macro- and micro-SAT manifest in a causal judgment task. As Experiment 1 did not address the sources of reasoning errors, we conducted a follow-up experiment, Experiment 2, which included confidence measures to elucidate the underlying processes responsible for reasoning errors.

## 2. Experiment 1

To test the effects of time pressure on causal probability judgments, we used an established causal inference task. In this task participants were asked to make judgments about events that are part of a known causal structure. This 'reasoning from a known structure' entails the applying knowledge of causal relationships to judge the probability of an event conditional on the state of other events in the structure. Besides the implementation of time pressure, the procedure and materials were based on multiple studies by Rehder and colleagues [32, 38, 62, 74]. The experiment was approved by the local ethics committee of the University of Amsterdam (nr. 2019-PML-10019).

### 2.1 Participants

41 individuals participated in the study for course credits (21 female, mean age 21.0) which took 41 minutes on average. All participants provided informed consent before participation in the study. We used an a priori exclusion criterion of an overall mean error above 18%. As the task uses a percentage response format, this criterion meant that participants with

responses on average more than 18 percentage points removed from the normative answer were excluded. This criterion was chosen before seeing the data and was based on the following logic: If a participant responds with '50%' on every single trial (a strategy clearly indicating non-compliance), they will have an average response error of 18%. This is due to the distribution of the normative responses (see green dashes in Fig 7), which are on average 18 percentage points removed from 50%. Hence, if a participant has an error lower than 18%, we can assume that on average they respond on the correct side of 50%, i.e. if the correct answer is higher (lower) than 50%, they tend to respond higher (lower) than 50%. This indicates compliance and understanding of the task. We cannot be sure of compliance or understanding for a participant with an average error larger than 18%. Such a participant could be systematically responding on the wrong side of 50%, that is, responding higher than 50% when the correct response is below 50% and vice versa. Therefore, using an exclusion criterion of 18% ensures that the participants included in the analysis did not engage in random responding or guessing. This led to the exclusion of 15 participants (see S1 Appendix for a supplementary analysis of these excluded participants). In addition, we removed responses faster than 1.5 seconds, which amounted to 1.8% of responses. In previous causal reasoning experiments, response times ranged between 5.5 and 23 seconds on average [18]. As each trial requires the processing and integration of five cues (three variable values and their causal relationships, see below), responses faster than 1.5 seconds are a clear indication of non-compliance.

## 2.2 Experimental design and procedure

The experiment was conducted in the behavioral sciences lab of the University of Amsterdam. The task consisted of three experimental domains, each consisting of a learning phase and a testing phase. In the learning phase participants learned a specific causal structure, about which they were asked to make inferences in the testing phase. Each testing phase consisted of three blocks with different response deadlines. Each of these blocks consisted of 27 trials. All participants completed 27 trials per domain and deadline condition, for a total of (27 x 3 x 3 =) 243 trials.

The three domains about which participants had to reason concerned meteorology, sociology, and economics (see Rehder, 2014). We tested three 3-variable causal structures (see Fig 1): a common cause, a chain, and a common effect structure. Each participant saw all three domains and all three causal structures. The order of the causal structures and which structure was paired with what domain was counterbalanced across participants (e.g. some participants had a common cause structure in the domain of economics, while others had a chain or common effect structure in the economics domain). The variables in the causal structures were binary, each with a "normal" and a non-normal value [38], which we will refer to in equations with 0 and 1 respectively. The non-normal value for each variable was either "high" or "low" and these values were counterbalanced across participants to control for effects of prior knowledge about the domains [18].

**2.2.1 Learning phase.** Each domain started with a learning phase. First, participants studied several computer screens with verbal information regarding the domain and how the variables are causally related in the specific causal structure. For example, participants that had the economics domain paired with a common effect structure, were taught that low interest rates (cause 1) and high trade deficits (cause 2) independently cause low retirement savings (effect). For the causal relationships it was always the case that non-normal values of variables caused the non-normal value of another variable. The causal structure and relationships was first described to participants in words (e.g.: "High interest rates cause small trade deficits"). For a complete description of all variables and causal relationships used see Appendix A in [18].

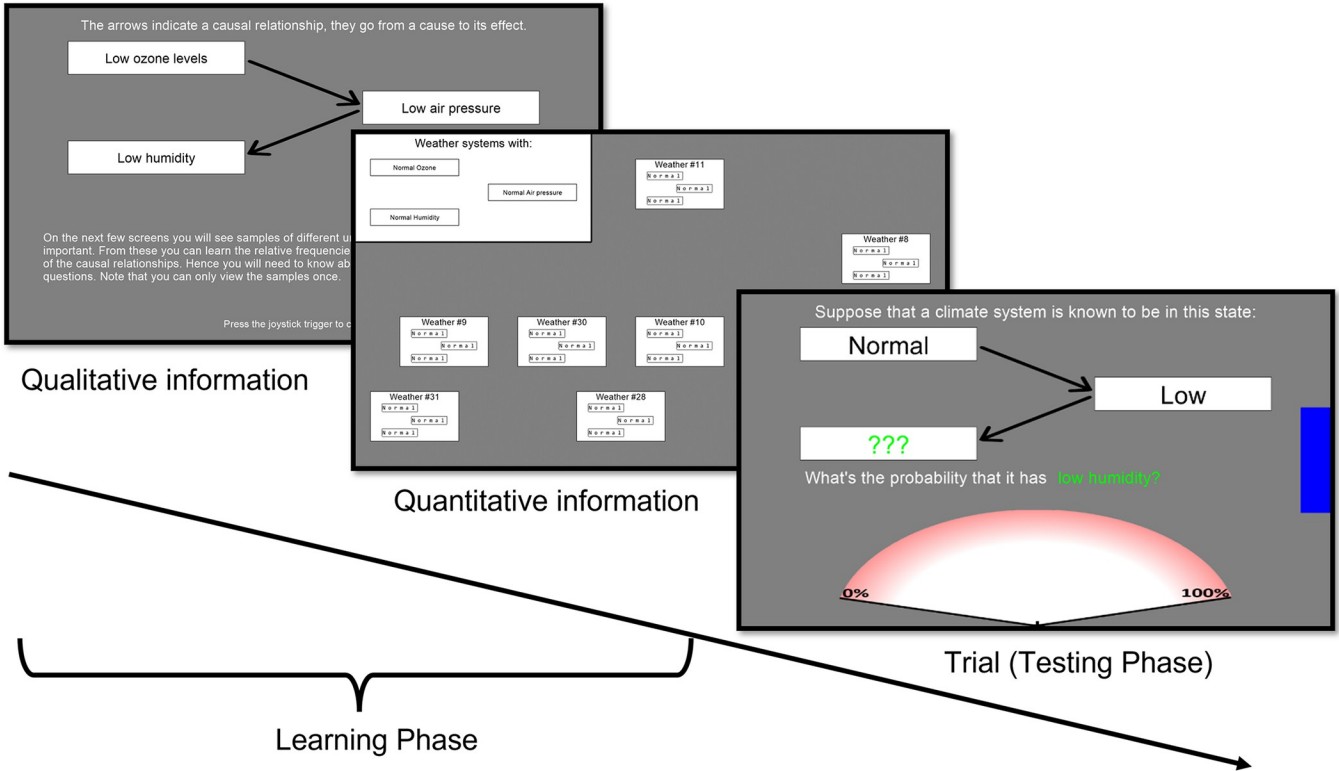

**Fig 2. Overview Experiment 1.** Overview and screenshots of one domain in Experiment 1. Participants start each domain by learning qualitative and quantitative information about the causal network (panels 1 and 2). The first panel shows a screenshot of how participants learned about the qualitative structure of the causal scenario. The second panel is a screen with learning samples that provided quantitative information. This was one of eight such screens participants saw in each learning phase. Here all variables had the 'normal' value (bottom row Table 1), on the 7 other screens the variables had different combinations of values. Each sample was numbered (e.g. 'Weather #23') to emphasize that they represented individual instantiations of the causal variables. The last panel is a trial in the meteorology domain. The blue bar on the right slowly decreased in size indicating the deadline. Participants had to judge the probability of the variable (here 'humidity') with the three green question marks being in its non-normal state (here 'low'). Participants moved a cursor from the center of the part-circle at the bottom of the screen over the edge of the part-circle to indicate their response.

After these descriptions participants viewed a graphical representation of the causal network, as in Fig 1, but with each node described as the relevant causal variable (e.g. "high interest rates" instead of "$X_1$", see Fig 2).

Next, participants received quantitative information by experience as has been done in previous studies [33, 38]. This method involves participants experiencing the causal relationships and their strengths by viewing case data, which is more comparable to how we learn causal information in daily life than to provide participants with written probabilities. For each of the eight possible combinations of variable values participants were presented with a separate screen showing a certain number of cases (each screen corresponded to one row in Table 1, see Fig 2 for an example of such a screen where all variables have a normal value). The quantitative information regarding the causal networks is learned by the relative number of cases for each possible combination of variable values. This method of teaching participants the network parametrization has been used successfully before [33, 38]. The network parametrization of the structures (and thus the number of cases on each sample screen) was taken from Experiment 1a in [33] and intended to be theoretically neutral. The chain and common cause structure had the same parametrization, with base rates of .5 for all variables, where we use 'base rate' to refer to the marginal probability of a variable across all cases, e.g. $P(X_1 = 1)$. The effects in the network had a probability of .75 when their parent cause was present and .25 when it

**Table 1. Parametrization of causal networks by cases.**

| Causal system state | | | Number of cases | |
|---|---|---|---|---|
| $X_1$ | Y | $X_2$ | Chain and common cause | Common effect |
| 1 | 1 | 1 | 9 | 6 |
| 1 | 1 | 0 | 3 | 4 |
| 1 | 0 | 1 | 1 | 2 |
| 1 | 0 | 0 | 3 | 4 |
| 0 | 1 | 1 | 3 | 4 |
| 0 | 1 | 0 | 1 | 0 |
| 0 | 0 | 1 | 3 | 4 |
| 0 | 0 | 0 | 9 | 8 |

*Note*. Parametrization of causal networks used in experiments, implemented as cases viewed by participants. $X_1$, Y, and $X_2$ refer to causal variables as presented in Fig 1. In the first three columns the 1s and 0s refer the to presence or absence respectively of a causal variable

was not. In the common effect structure, the two causes combined by way of a Noisy-OR gate [75] with causal strengths of .5 and with base rates of .5. This meant that the effect had 0 probability if no causes were present, .5 when one cause was present, and .75 when both causes were present (hence the base rate was .43 for the effect). This parametrization was shown as cases on the sample screens according to Table 1. The CBN predictions, i.e. the correct response, for each inference is completely determined by this parametrization and obtained using probability calculus [7].

**2.2.2 Testing phase.** The testing phase immediately followed each learning phase of a particular domain. Participants judged the probability of a specific variable being in their non-normal state (e.g. "what is the probability of retirement savings being low?"), while the other two variables are presented as in one of three states: unknown, or having their normal or non-normal value (e.g. "trade deficits are normal and interest rates are low", see Fig 2). We asked all (3 choices for the queried variable x 3 possible states of first conditional variable x 3 possible states of second conditional variable =) 27 possible questions three times, each under a different level of time pressure, resulting in 81 inferences per domain.

The testing phase of each domain was split up into three blocks which had response deadlines of 6, 9, and 20 seconds (henceforth referred to as DL6, DL9, and DL20). The choice of deadlines was based on pilot studies, given the lack of experimental findings on the effect of deadlines on causal probabilistic judgments. The presentation order of the blocks was counterbalanced across domains. At the start of each block a screen indicated the response deadline for the next 27 trials. On each trial a blue bar on the right indicated how much time was left to respond (Fig 2). When participants failed to answer before the deadline, which happened on a total of 19 trials over all participants, a screen was presented for 5 seconds before starting the next trial that told participants that they had to respond as accurately as possible while not missing the deadline.

Participants responded on a probability scale ranging from 0% to 100% using a joystick. This setup enabled fast, intuitive responses and reduced variance in RTs due to response execution (see for other uses of joysticks in psychological experiments [76, 77], and for the validity of using a joystick for continuous responses see [78]). The 0%-100% range was presented on the edge of part of a circle around the starting point of the cursor controlled by the joystick (see Fig 2, bottom of third panel). Participants were instructed to move the joystick in one swift movement so that the cursor would cross the edge of the circle at the location responding to their answer.

**2.2.3 Analyses.** For our main analyses we applied mixed-effects regression models using the lme4 package in R [79]. As predictors we included the theoretically relevant variables and their interactions, i.e. RT and deadline condition for all regressions, and for analyses testing specific errors this included variables indicating the state of the causal variables relevant for that error. We included crossed random intercepts for participants and the different inferences [80].

We used linear regressions for RTs and probability judgments, where the judgments were rounded to percentage points. The response error of a judgment was defined as the absolute difference between a response and the normative answer (i.e., lower values indicate a less error). Response error is strictly non-negative and positively skewed, and so we used a Gamma distribution with a log link function for the regressions on error. Because of this, for the regressions on error, the effects need to be interpreted multiplicatively (instead of additively in standard regression). We added 0.01 to all observations to avoid responses with an error of 0 (there were 13 such responses in total), as the Gamma distribution is only defined for positive values.

Both probability judgments and response error are coded on a percentage scale (from 0 to 100), and RT is z-scored within participants. As the common cause and chain structures have an identical normative joint distribution (see Table 1), both response error and the systematic reasoning errors are measured in exactly the same way for these structure (e.g. the inferences relevant to Markov violations are exactly the same). Hence, we analyze these structures together.

We report estimates and statistics from regression models with insignificant higher-order interactions removed. To test for this significance of effects we use Satterthwaite adjusted F-tests for the linear mixed regression models [81]. Where we applied a Gamma regression for non-normally distributed dependent variables, we use Wald Chi-square tests. Where relevant we also report estimated marginal means, i.e. estimations of the dependent variable based on the regression model for a predictors of interest while averaging over other predictors in the model. Post-hoc contrasts will be reported with p-values adjusted for multiple comparisons using the Tukey method.

## 2.3 Results

The Results section is structured as follows. We first check whether the deadline manipulation was effective, next we look at the effects of time pressure on overall error (overall SAT), and then we investigate the effects of time pressure on the three systematic reasoning errors (Markov violations, failures to explain away, and lastly, conservatism).

**2.3.1 Manipulation check.** To test whether the time pressure manipulation impacted response times we regressed the Deadline factor on RTs and found that the effect of Deadline is indeed significant ($F(2, 6153) = 668$, $p < .001$, Fig 3). This is reassuring considering the lack of existing information on time pressure manipulations in this domain.

**2.3.2 Overall response error.** Next, we investigated the overall SAT, that is, the influence of RT and deadline conditions on overall error. We found a significant main effect of Deadline ($\chi^2(2) = 23.8$, $p < .001$, Fig 4A), indicating a macro-SAT (Fig 4A): Participants were more accurate when there was less time pressure. Post-hoc contrasts revealed that this is due to participants having less response error in the DL20 condition ($M = 12.6$, $SE = 0.680$, $z_{\text{DL6-DL20}} = 4.56$, $p < 0.001$, $z_{\text{DL9-DL20}} = 3.54$, $p = .0012$), while we do not find a difference between the DL6 ($M = 14.5$, $SE = 0.818$) and DL9 ($M = 13.8$, $SE = 0.737$) conditions ($z = 1.73$, $p = .195$). We also found a significant interaction effect of RT and Deadline ($\chi^2(2) = 8.84$, $p = .012$, Fig 4B), revealing a micro-SAT. While the interaction indicated that the effect of RT is less strong with

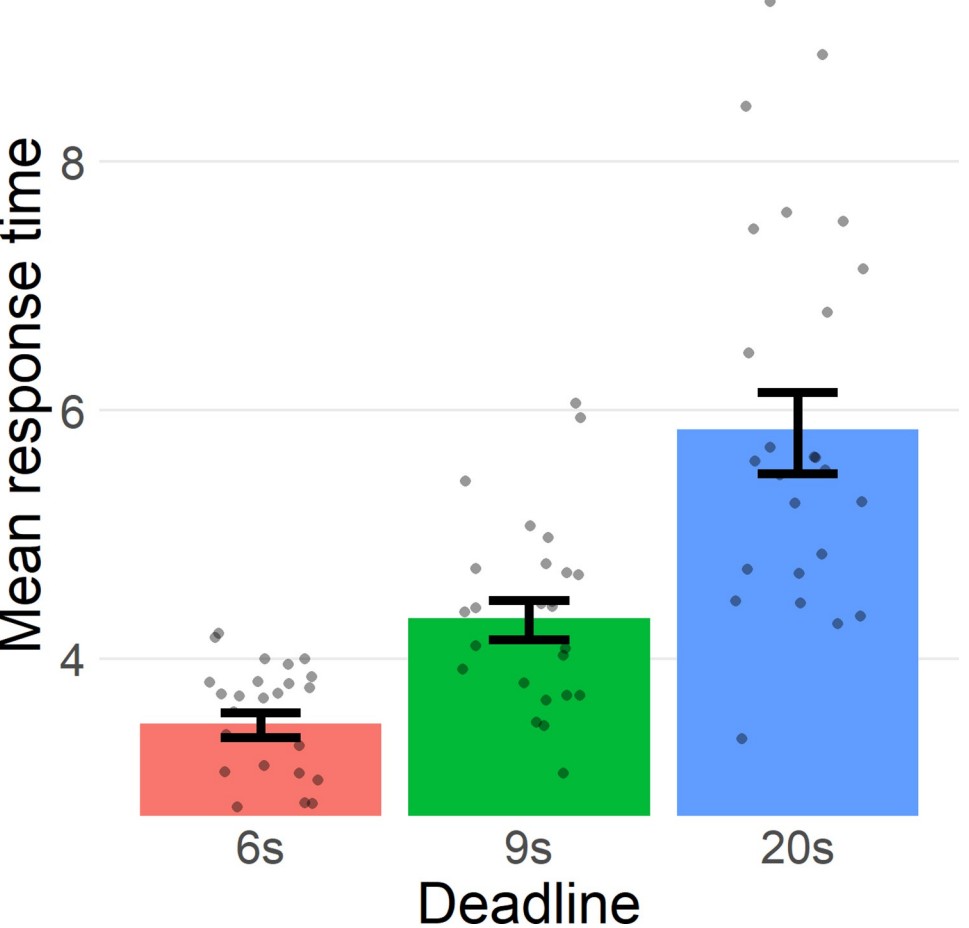

**Fig 3. Mean response times per deadline.** Mean response times per deadline condition. Dots represent the mean response times of each participant. Bars indicate standard error of the mean.

longer deadlines, we found that the effect of RT is significant for each of the deadlines (DL6: $M = 1.16$, $SE = 0.0432$, $z = 3.72$, $p < .001$; DL9: $M = 1.07$, $SE = 0.0241$, $z = 3.09$, $p = .002$; DL20: $M = 1.03$, $SE = 0.0131$, $z = 2.43$, $p = .015$). This means that participants were less accurate the closer they got to the deadline, and that this effect was strongest for the shorter deadlines. Notably, the micro-SAT is in the opposite direction of the macro-SAT: within deadline conditions a longer RT is associated with being less accurate (micro-SAT), while over deadline conditions we see that people are most accurate in the conditions with longer RTs (macro-SAT). In other words, the direction of the relationship between RT and error depends on whether one looks within or across deadline conditions.

**2.3.3 Markov violations and explaining away.** To test the effect of the deadlines and RT on Markov violations we performed more mixed model regressions. Here we analyzed only those inferences relevant to Markov independence, that is, inferences about a terminal variable ($X_1$ or $X_2$), while the middle variable was known (either $Y = 0$ or $Y = 1$) for the common cause and chain structures, or while the middle variable was unknown for the common effect structure. The dependent variable was the response in percentage points. We included a variable as fixed effect indicating whether the other terminal event (the screened off variable, $X_2$ or $X_1$) was absent, unknown, or present (coded as -1, 0, 1). A significant effect of this factor (henceforth referred to as ScreenedOff) thus indicates a violation of Markov independence.

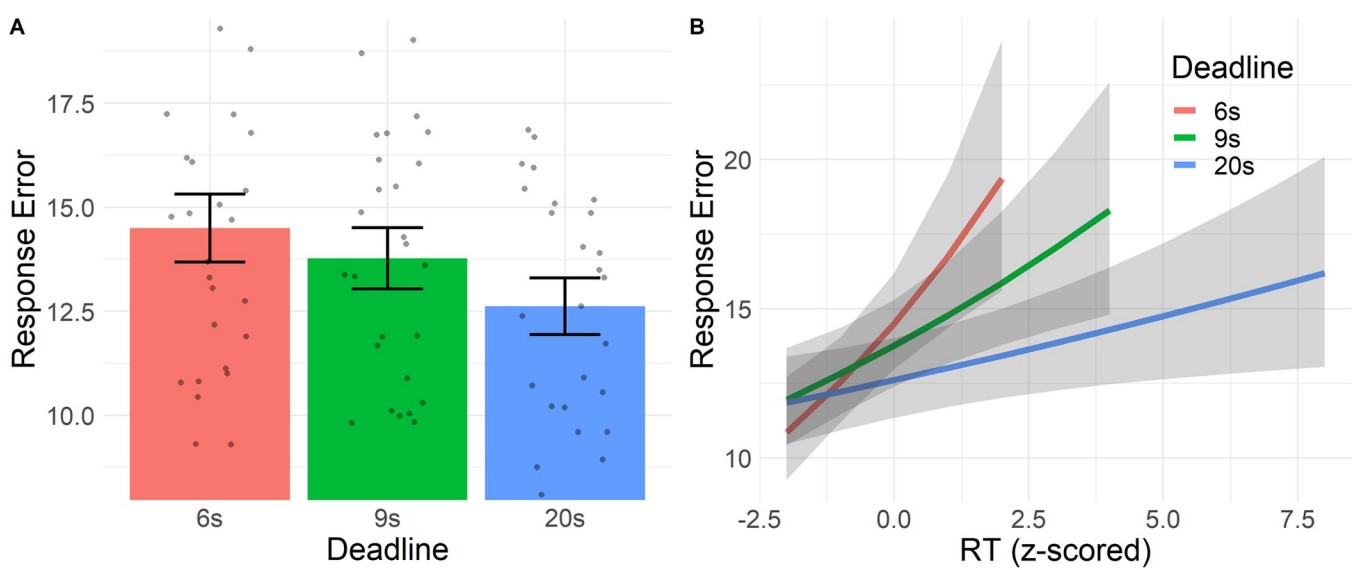

**Fig 4. Overall response error.** Estimated response error from mixed-effects regression. A. Error in each deadline condition. Error is on the Y-axis, defined as the absolute difference in percentage points between response and normative answer (hence, lower values indicate that participants are more accurate). Dots indicate mean error for each participant. The bars represent standard errors. B. Interaction effect of deadline and RT on error. Error is on the Y-axis, Z-scored RTs are on the X-axis. The lines are estimated marginal means, the ribbons represent the 95% confidence interval.

Additional fixed effects for the status of the middle variable (values: Y = 0 or Y = 1, MidVar, only for the common cause and chain structures), RT, Deadline, and their interactions with ScreenedOff were included in the model.

*2.3.3.1 Markov violations in common cause and chain structures.* As expected, we found a significant main effect of ScreenedOff ($F(2, 1800) = 128$, $p < .001$), indicating that participants did not screen off, and thus violated Markov independence. The interactions of ScreenedOff with Deadline ($F(4, 1801) = 0.641$, $p = .633$) and RT ($F(2, 1809) = 1.68$, $p = .186$) were both not significant, indicating that the violations of Markov independence were not impacted by time pressure nor response times (Fig 5A and 5B). While these results (and Fig 5) are rather convincing, the lack of a significant effect does not provide direct evidence for the absence of such an effect. As this result is surprising and important for our aims, we additionally computed Bayes factors for the effects of time pressure and RT on reasoning violations and conservatism, both here and in subsequent sections. Bayes factors provide strong evidence against an effect of both deadlines ($BF_{01} > 100$) and RT ($BF_{01} = 23.0$). We did find a significant interaction between ScreenedOff and MidVar ($F(2, 1801) = 15.7$, $p < .001$)), indicating that the violations of Markov dependence were larger when the middle variable was present than when it was not (Fig 5A and 5B).

*2.3.3.2 Markov violations in common effect structure.* We performed a similar analysis for the common effect structure, the only difference being that the model did not include the variable MidVar. Hence the initial model included fixed effects for ScreenedOff, Deadline, RT, and the interactions of the latter two with ScreenedOff. We again only found a significant main effect of ScreenedOff ($F(2, 415) = 29.4$, $p < .001$), indicating that participants violated Markov independence here. The interactions with ScreenedOff were not significant for both Deadline ($F(4, 415) = 1.65$, $p = .161$, $BF_{01} = 5.85$) and RT factors ($F(2, 419) = 0.0105$, $p = .9896$, $BF_{01} = 24.0$, Fig 5C).

*2.3.3.3 Explaining away.* To test the effect of the deadlines and RT on explaining away, we conducted another mixed model regression using only the data from inferences relevant to

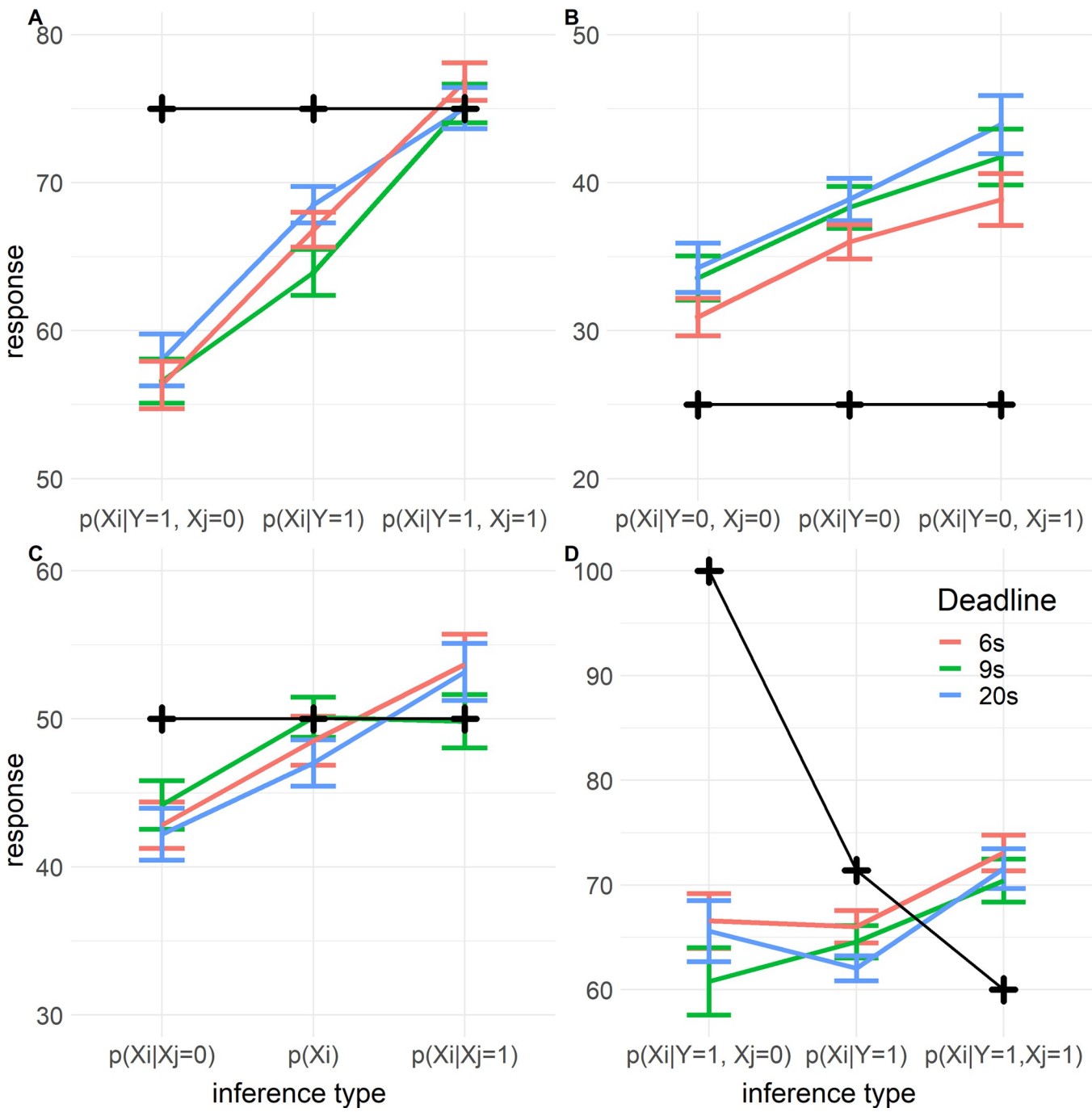

**Fig 5. Markov violations and explaining away per deadline in Experiment 1.** Y-axis indicates response on a percentage scale. Colored lines indicate mean responses, the error bars indicate their standard errors. The black crosses indicate the normative response. The X-axis indicates the specific inference. Symmetric inferences are collapsed, e.g. 'Xi | Xj = 0' refers to both 'P(X_1 = 1| X_2 = 0)' and 'P(X_2 = 1| X_1 = 0)'. A. Markov violations in common cause and chain structures where the middle variable is present (Y = 1). B. Markov violations in common cause and chain structures where the middle variable is absent (Y = 0). C. Markov violations in common effect structure. D. Explaining away in common effect structure.

explaining away. That is, those inferences in the common effect structure about one of the causes, while knowing that the effect (the middle variable) is present. We computed a variable (AwayVar) indexing whether the non-queried terminal event ($X_j$ above) was absent, unknown,

or present (coded as -1, 0, 1 respectively). Next, we computed a new dependent variable by recoding participants' responses on these trials such that in our regression model the effect of AwayVar would be zero if participants respond correctly on each trial. The correct responses for $P(X_i = 1 | Y = 1, X_j = 0)$, $P(X_i = 1 | Y = 1)$, and $P(X_i = 1 | Y = 1, X_j = 1)$ are 1, .714, and .6 respectively. To compute the recoded dependent variable we subtracted .286 from responses to the inference where $X_j = 0$, and added .114 to the responses on trials where $X_j = 1$. We did this such that for this recoded response variable the correct responses for the explaining away inferences are all 0.714 (since $1 - 0.286 = 0.714$, and $0.6 + 0.114 = 0.714$). This means that we can interpret the effect of AwayVar as deviations from the normative pattern of explaining away. Our model included AwayVar, Deadline, RT, and the interactions of Deadline and RT with AwayVar.

We found a significant main effect of AwayVar ($F(2, 426) = 432$, $p < .001$), indicating that participants did not engage in the normative explaining away pattern (clearly visible in Fig 5D). When participants learn that the non-queried cause is absent compared to when it is unknown, participants should increase their response by 28.6% (i.e. $P(X_i = 1 | Y = 1, X_j = 0) - P(X_i = 1 | Y = 1) = .286$), see black line in Fig 5D). However, post-hoc contrasts reveal that participants only increased their response by 0.46% ($SE = 2.80$) on average. Similarly, when learning that the non-queried cause is present compared to it being unknown, they should decrease their estimate by 11.4% ($P(X_i = 1 | Y = 1, X_j = 1) - P(X_i = 1 | Y = 1) = -.114$), but participants only changed their judgement with +4.88% ($SE = 2.89$). In sum, participants failed to properly 'explain away'.

Next, let us regard the effects of time pressure on explaining away. There was no influence of the deadlines on how participants explained away ($F(4, 426) = 1.27$, $p = .280$, $BF_{01} = 11.0$). However, we did find mixed evidence of an interaction of AwayVar with RT ($F(2, 433) = 3.64$, $p = .0270$, $BF_{10} = 1.09$). We plotted the estimated interaction in Fig 6. From this interaction we can see that RT impacted responses on trials where the non-queried cause is present ($t(435) = -3.36$, $p < .001$), while RT had no effect when the non-queried cause is absent ($t(432) = -0.79$, $p = .43$), or when its status is unknown ($t(434) = -0.21$, $p = .83$). The responses on trials where the non-queried cause is present got closer to 50% percent as participants took longer to respond. It is possible that this effect is not related to explaining away, but rather to conservative inferences. The inference where the RT has an effect is where responses are most extreme and so we would expect conservatism to be more pronounced for this inference. We return to this in the discussion.

**2.3.4 Conservatism.** We now look at the last persistent reasoning error, namely the tendency for reasoners to be conservative. Participants tended to respond conservatively with responses being on average 4.9 percentage points ($SE = 0.84$, $t = 5.85$, $p < 0.001$) closer to 50% than the normative response (see Fig 7). To quantify the relationship between time pressure and conservative responses, we computed a variable that measured the distance that a response moved from the normative answer towards 50% for all conflict (e.g. $P(Y = 1 | X_i = 1, X_j = 0)$) and ambiguous (e.g. $P(Y = 1 | X_i = 1)$) inferences. Positive values for this variable indicate that a response was in between 50% and the normative answer (or at 50%). Positive values thus indicate conservative inferences. Negative values indicate that a response was more extreme (closer to 0% or 100%) than the normative response. Because this variable cannot represent responses for which the normative response is exactly 50% these trials were excluded from this analysis (21.9% of all trials). Additionally, we removed responses that were on the opposite side of 50% from the normative response (12.6% of all trials). For example, if the normative response was larger than 50%, but a participant gave a response below 50%, that trial was removed.

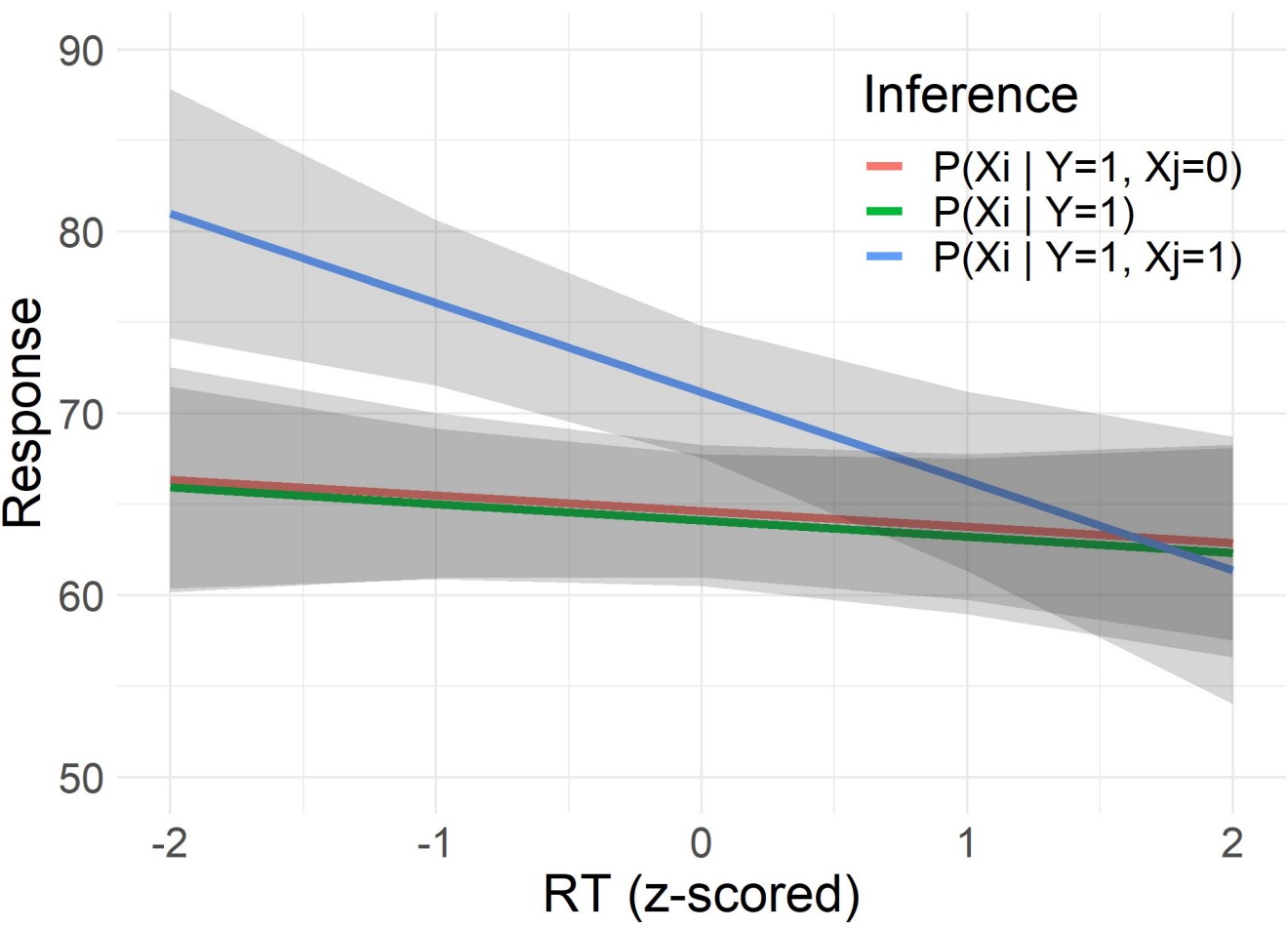

**Fig 6. Interaction of explaining away with response time.** Estimated responses from mixed-effects regression in common effect structure based on RT and AwayVar in Experiment 1. This plot visualizes the interaction effect AwayVar x RT on responses, i.e. the interaction between RT and explaining away, where horizontal lines would indicate no effect of RT and differently sloped lines indicate an interaction. AwayVar here refers to the status of the non-queried cause in the inference on the common effect structure, where -1 indicates the non-queried cause is absent, 0 that its status is unknown, and 1 that the non-queried cause is present. The Y-axis indicates responses on percentage scale, the X-axis indicates RT (z-scored). The grey ribbons represent 95% confidence intervals.

To test the impact of time pressure on conservatism, we employed a regression model using the metric of conservative responding as dependent variable. We found a significant interaction effect of Deadline and RT on conservatism ($F(2, 2673) = 6.89$, $p = .001$, $BF_{10} = 6.48$, Fig 8). Using post-hoc contrasts, we found that the effect of RT is significant in the 6s ($\beta = 1.90$, $SE = 0.679$, $t(2676) = 2.79$, $p = .0054$) and 9s deadlines ($\beta = 1.49$, $SE = 0.378$, $t(2672) = 3.95$, $p < 0.001$), but not for the 20s deadline ($\beta = 0.132$, $SE = 0.215$, $t(2672) = 0.612$, $p = .54$). Pairwise contrasts revealed that the effects in the 6s and 9s conditions are not significantly different ($t(2672) = 0.520$, $p = .86$), while they were different from the 20s condition (versus 6s: $t(2676) = 2.48$, $p = .036$; versus 9s: $t(2672) = 3.14$, $p = .0049$). Hence there seemed to be a micro-SAT for conservative inferences in the 6s and 9s conditions, but not in the 20s condition. This is in line with a main effect of RT ($F(2, 2678) = 18.5$, $p < .001$, $BF_{10} = 4.59$). We found mixed evidence for a main effect of Deadline ($F(2, 2668) = 4.40$, $p = .012$, $BF_{01} = 9.34$) which could indicate a macro-SAT. Contrasts indicate that there is more conservatism in the 6s deadline condition ($M = 6.39$, $SE = 1.43$) compared to the deadlines of 9s ($M = 4.93$, $SE = 1.39$, $t(2670) = 2.74$, $p = .017$) and 20s ($M = 4.95$, $SE = 1.39$, $t(2670) = 2.67$, $p = .020$).

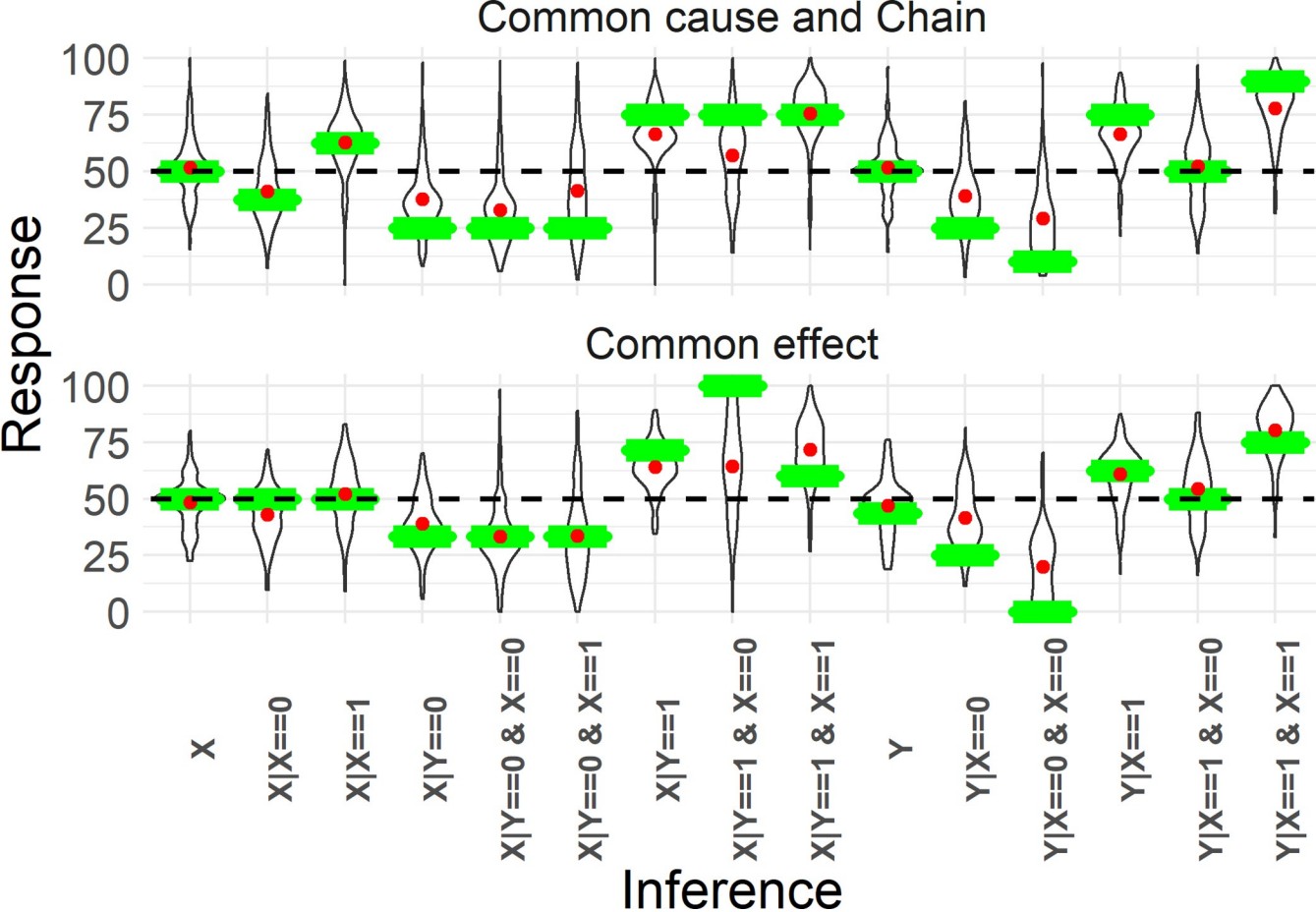

**Fig 7. Mean responses per inference.** This figure indicates conservatism in both Common cause and Chain (top) and Common effect structures (bottom). Y-axis represents the response (in %), X-axis indicates the type of inference. We collapsed over the symmetry between $X_1$ and $X_2$. The violin plots indicate the response distribution of all participants, the red dot is the mean response. The green bars indicate the normative (CBN) response. The horizontal dashed black line indicates responses at 50%. Conservatism can be seen by red dots that are closer to 50% than the green bars (for inferences where the normative response is not 50%).

## 2.4 Discussion

Taken together, the results from Experiment 1 indicate that there is an overall macro-SAT in causal probability judgements. Time pressure decreases the accuracy of responses as compared to the normative CBN point prediction. In addition, we found evidence for a micro-SAT. Responses with longer RTs are generally less accurate, and this micro-SAT is stronger in the conditions with more time pressure. The micro- and macro-SATs are in opposite directions, meaning that whether longer RTs are associated with improved accuracy depends on whether one looks within or across conditions [64].

With regard to the systematic reasoning errors, the results of Experiment 1 reveal that Markov violations and failures to explain away are not impacted by time pressure. Of the inferences relevant for explaining away, the only effect of time pressure we found was an effect of RT on the inference where the non-queried cause is present (Fig 6). However, it seems that this is due to conservative responding, as we do not see the effect on the other inferences relevant for explaining away. For the single inference where we find an effect of RT, participants provided estimates closer to 50% when RTs were longer. This could indicate that participants

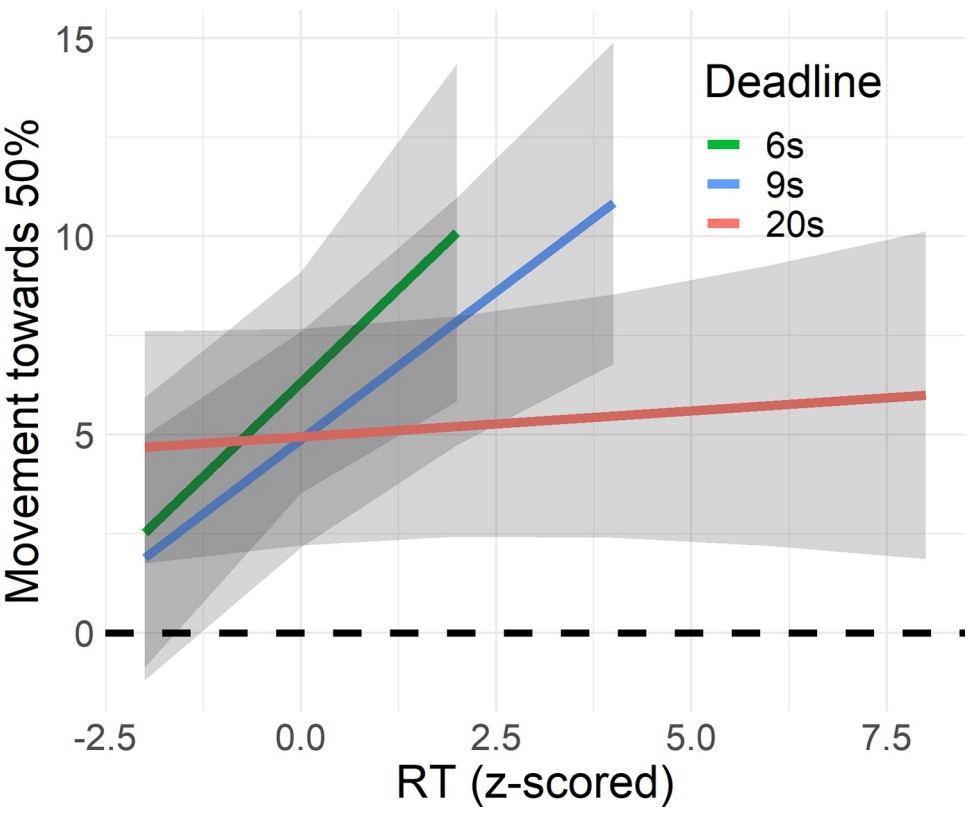

**Fig 8. Effect of deadlines and response time on conservatism.** Estimated movement towards 50% based mixed-effects regression in Experiment 1. This plot visualizes the Deadline x RT interaction on conservatism. The Y-axis represents conservatism, that is, the distance a response moved from the normative answer towards 50%. The X-axis represents RT (z-scored). The colored lines indicate the predictions, separated per deadline. The grey ribbons represent 95% confidence intervals.

grasped the idea that the non-queried cause being present should not increase the probability of the queried cause when they took more time to respond, i.e. participants grasped explaining away when they took more time. If this were the case, however, the question remains why we do not see the opposite effect of RT for those trials where the non-queried cause is absent, which we would expect if the effect of RT is related to grasping the idea of explaining away. Therefore, a more plausible explanation for the effect of RT is not related to explaining away, but to conservatism. If the passing of time affects conservative responding, we would most clearly see this on trials where responses are farther away from 50% and this is what we find here.

This conjecture is bolstered by the fact that we do find that conservative responding is impacted by time: Both a shorter deadline as well as the passing of time increase conservative responding. This latter effect is strongest in the conditions with short deadlines. This is the same pattern found for overall response error, and so it seems that changes in accuracy associated with time pressure and RTs are due to changes in conservative responding.

The observation that conservative responding is differently related to time pressure than Markov violations and failures to explain away is an indication of different cognitive processes. In particular, an interesting hypothesis is that the conservatism is the result of an increased probability to respond at or near 50% when time increases, effectively guessing [82]. This would entail that slow responses are associated with low confidence, reflecting that participants

were unsure of those answers [83]. Hence conservative responding might not be an error specific to causal reasoning, but the result of a more general cognitive principle related to uncertainty. This hypothesis is tested in Experiment 2.

## 3. Experiment 2

Experiment 2 tests the hypothesis that the increase in conservative inferences under time pressure is related to a decrease in confidence. In addition, Experiment 2 serves as a replication of Experiment 1, which seems opportune given the scarcity of experimental findings on time pressure effects in causal judgments. Moreover, there was a sizeable dropout rate (37% of participants) in Experiment 1 due to the a priori exclusion criterion we set on the overall error of responses (see S1 Appendix for an analysis of the excluded participants). Such a sizable dropout is not uncommon for demanding causal reasoning tasks like ours. For example, in a set of experiments on causal attribution the dropout rate ranged from 29% to 44% [84], in a set of studies on diagnostic inference it has been consistently around 30% [16, 85], and a set of experiments on the effect of prescriptive norms on causal inferences had dropout rates of up to 37% [86]. But, while a dropout rate like in Experiment 1 is not uncommon, it still behooves us to replicate our findings.

### 3.1 Confidence

Besides replication another goal for Experiment 2 was to study the relationship between confidence and conservatism. Confidence has been considered an important component of reasoning and decision-making more generally [83, 87–90]. Confidence tends to correlate negatively with RT, and higher confidence is associated with more accurate judgments [83]. An important consequence of employing time pressure manipulations is that participants need to make responses with varying levels of confidence [88, 91, 92], which we also expect to observe. With regard to causal reasoning, one recent study has shown that confidence can predict verbal causal ratings [93], indicating that we can expect confidence to be relevant in understanding causal judgments. Additionally, confidence has been used to test theories of memory retrieval [89, 94], sensory processing [95], and decision-making [96, 97], suggesting that confidence ratings can indeed reflect differences in cognitive processing.

### 3.2 Methods

**3.2.1 Participants.**   37 individuals participated in the study (9 female, mean age 28.3) for a monetary reward of £5.63 via the Prolific platform (www.prolific.co). We selected participants that had a 100% approval rating for previous studies they participated in on Prolific. All participants provided informed consent and the experiment took around one hour to complete. We used the same exclusion criteria as in Experiment 1. This resulted in the exclusion of 20 participants due to a mean error above 18% (we return to this in the discussion, see also S1 Appendix for an analysis of the excluded participants), and the removal of 2.2% of responses with an RT of lower than 1.5 seconds.

**3.2.2 Design and procedure.**   The experimental design of Experiment 2 was almost identical to Experiment 1, but differed from it in three ways: (1) the experiment was conducted online rather than in a physical laboratory, (2) participants responded using their mouse or trackpad instead of a joystick, and (3) at the end of each trial participants reported their confidence. The use of a mouse or trackpad required an additional screen after each response where participants were presented with their current cursor position and were asked to move it back to the middle of the screen. The crucial difference between the experiments was that, in addition to participants' probability estimates, we now also asked participants after each trial to

report the confidence they had in their response. Identical to the probability judgments, participants moved a cursor over the edge of part of a circle, which here ranged from 'not confident at all' to 'completely confident'.

**3.2.3 Analyses.** We used the same mixed effects regression approach as in Experiment 1. For the analyses regarding confidence we added z-scored confidence reports as an additional predictor. Data and analysis code are publicly available at https://osf.io/bz9vj/.

## 3.3 Results

**3.3.1 Replication of findings from Experiment 1.** We largely replicated the effects on overal response error and the systematic reasoning errors from Experiment 1. Hence, we only briefly report here the main results related to the systematic reasoning errors (additional analyses are reported in S2 Appendix).

*3.3.1.1 Markov violations and explaining away.* We found no effect of deadlines ($F(4, 1171)$ = 0.960, $p$ = .43, $BF_{01}$ = 48, Fig 9A and 9B) or RT ($F(2, 1180)$ = 2.73, $p$ = .065, $BF_{01}$ = 3.13) on Markov independence violations in common cause and chain structures, nor did we find an effect on these violations in the common effect structure (Deadline: $F(4, 262)$ = 1.08, $p$ = .37, $BF_{01}$ = 9.93, Fig 9C; RT: $F(2, 265)$ = 1.73, $p$ = .18, $BF_{01}$ = 3.89). For explaining away we found no effect of deadlines ($F(4, 271)$ = 1.18, $p$ = .32, $BF_{01}$ = 11.7, Fig 9D), but similar to Experiment 1 we found an indication of an effect of RT ($F(2, 279)$ = 6.29, $p$ = .002, $BF_{10}$ = 0.986). This latter effect seems again to be due to conservative responding as in Experiment 1. We found that the effect of RT is significant for the inferences where the non-queried cause is present ($P(X_i = 1 | Y = 1, X_j = 1)$, β = -6.58, $SE$ = 1.44, $t(280)$ = -4.56, $p$ < .001) or unknown ($P(X_i = 1 | Y = 1)$, β = -4.30, $SE$ = 1.64, $t(275)$ = -2.63, $p$ = .009). For these inferences participants responded closer to 50% when RTs were longer. There was no effect of RT on the inferences where the non-queried cause is absent ($P(X_i = 1 | Y = 1, X_j = 0)$, β = 1.77, $SE$ = 1.57, $t(276)$ = 1.12, $p$ = .263).

*3.3.1.2 Conservatism.* Lastly, we found mixed evidence of an interaction of Deadline and RTs on conservative responding ($F(2,1676)$ = 4.55, $p$ = .011, $BF_{10}$ = 0.739). Focusing on main effects, we find that there is no effect of Deadline on conservatism ($F(2,1673)$ = 1.93, $p$ = .15, $BF_{01}$ = 21.3), but we find a large effect of RT ($F(1,1681)$ = 21.5, $p$ < .001, $BF_{10}$ > 100) indicating that conservatism is sensitive to internal time pressure.

Having replicated the main findings of Experiment 1, let us now look at confidence as a predictor of overall response error and systematic reasoning errors.

**3.3.2 Confidence and overall response error.** Regression analysis showed there was a main effect of confidence on error ($\chi^2(1)$ = 24.0, $p$ < 0.001, β = -6.14, $SE$ = 1.25), indicating that more accurate responses were associated with higher confidence. This result gives credence to the use of confidence as an index of participant's uncertainty about their inference. We also found an interaction effect of confidence and RT on response error ($\chi^2(1)$ = 5.15, $p$ = .023, β = -2.62, $SE$ = 1.16). For responses associated with low confidence longer RTs imply larger errors, while for responses with high confidence longer RTs imply smaller errors. The most likely interpretation for this finding is that we find longer RTs for two reasons. Sometimes long RTs reflect more deliberation, leading to more accurate responses and higher confidence. But sometimes long RTs reflect that the problem is difficult, resulting in less accurate and less confident responses. There was no interaction of confidence with the deadlines ($\chi^2(2)$ = 0.822, $p$ = .66). In addition, the main effect of Deadline on overall error remains significant ($\chi^2(2)$ = 6.47, $p$ = .0394).

**3.3.3 Confidence in Markov violations and explaining away.** To test the role of confidence in Markov violations and explaining away, we included confidence as an additional predictor in our regression analyses and inspected its interactions with the violations.

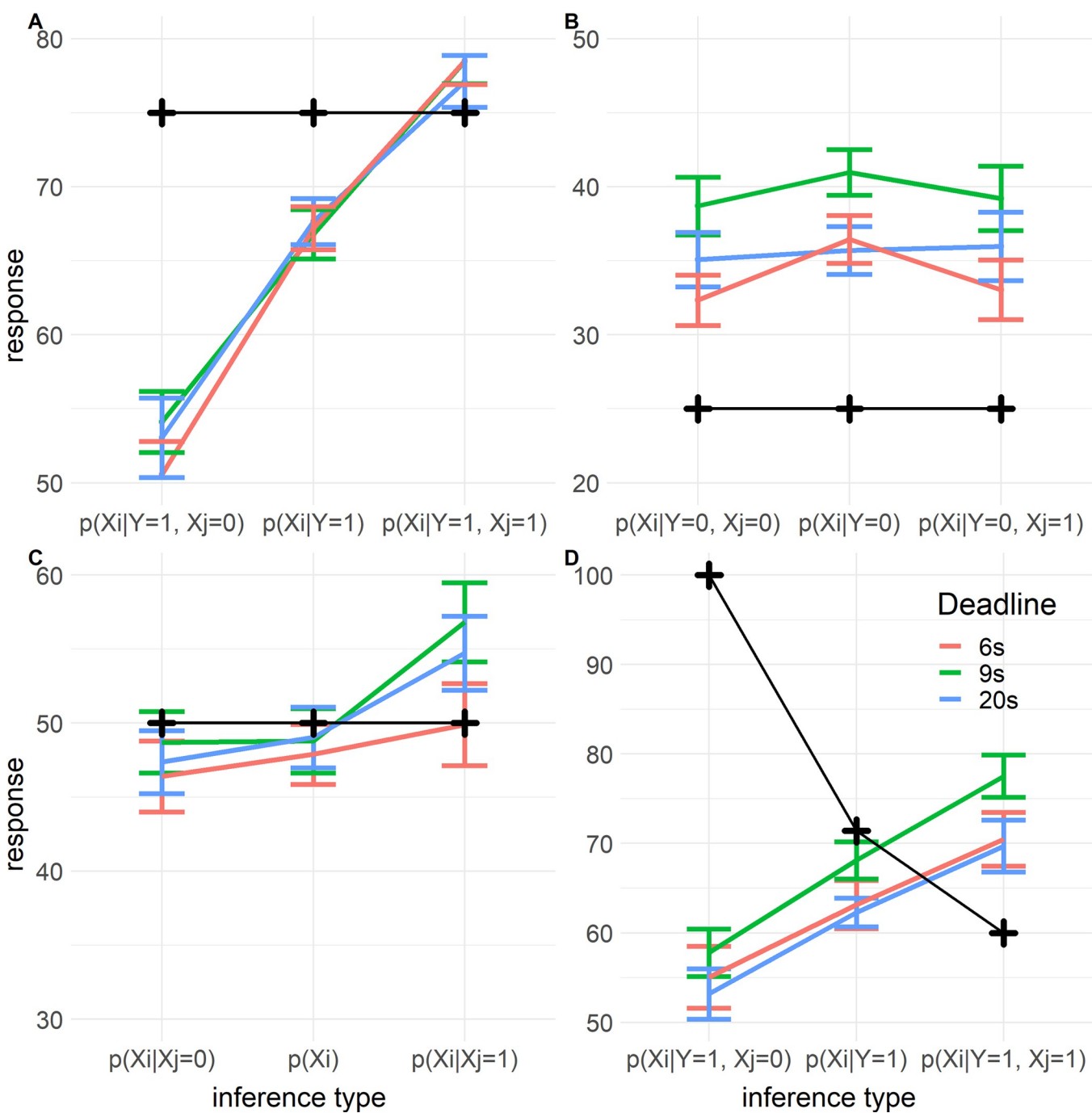

**Fig 9. Markov violations and explaining away per deadline in Experiment 2.** Markov violations and explaining away per deadline in Experiment 2. Y-axis indicates response on a percentage scale. Colored lines indicate mean responses, the error bars indicate their standard errors. The black crosses indicate the normative response. The X-axis indicates the specific inference. Symmetric inferences are collapsed, e.g. 'Xi | Xj = = 0' refers to both 'X$_1$ | X$_2$ = = 0' and 'X$_2$ | X$_1$ = = 0'. A. Markov violations in common cause and chain structures where the middle variable is present (Y = 1). B. Markov violations in common cause and chain structures where the middle variable is absent (Y = 0). C. Markov violations in common effect structure. D. Explaining away in common effect structure.

*3.3.3.1 Common cause and chain structures.* For the common cause and chain structures we find a significant three-way interaction effect of confidence with the screened off variable (ScreenedOff) and the status of the middle variable (MidVar; $F(2, 1160) = 6.21$, $p = .003$, $BF_{10}$

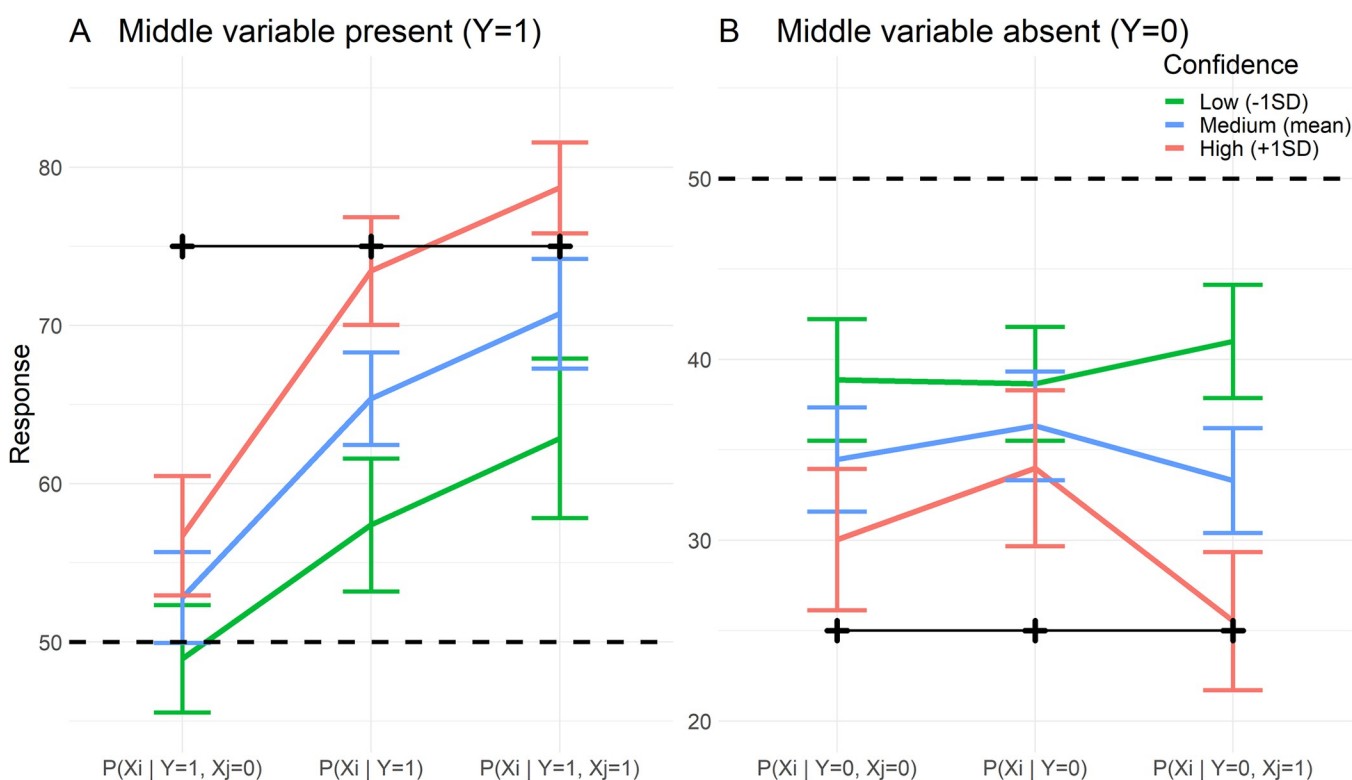

**Fig 10. Effect of confidence on Markov violations.** Effect of confidence on Markov violations in common cause and chain structures. Colored lines are estimated responses based on regression model, error bars indicate 95% confidence interval. The estimates are based on different levels of confidence, low (-1$SD$), medium (mean), and high (+1$SD$). Black crosses and solid black lines indicate normative answers. Dashed line indicates 50%. This plot visualizes the three-way interaction effect of confidence with the screened off variable (ScreenedOff) and the status of the middle variable (MidVar).

= 1.99) as well as a significant two-way interaction of confidence with ScreenedOff ($F$(2, 1164) = 4.13, $p$ = .016, $BF_{10}$ = 4.78). To understand the relationship between these Markov violations and confidence, we plotted the response estimates at different levels of confidence in Fig 10. Firstly, post-hoc testing indicates that there is no violation of Markov independence when the middle variable is absent (Y = 0), as there are no differences between the levels of ScreenedOff ($\Delta_{\text{ScreenedOff -1 vs 0}}$ = 1.86, $SE$ = 1.48, $t$(1155) = 1.26, $p$ = .419; $\Delta_{\text{ScreenedOff 0 vs 1}}$ = 3.04, $SE$ = 1.49, $t$ (1155) = 2.04, $p$ = .103). In the case when the middle variable was present (Y = 1), there were significant Markov violations ($\Delta_{\text{ScreenedOff -1 vs 0}}$ = 12.6, $SE$ = 1.43, $t$(1155) = 8.23, $p < .001$; $\Delta_{\text{ScreenedOff 0 vs 1}}$ = -5.36, $SE$ = 1.74, $t$(1159) = -3.09, $p$ = .0059). For this case where the middle variable was present, we can see from the slopes of the colored lines in Fig 10A that the Markov violations were stronger when confidence is high (the red lines are the steepest) when comparing the inferences where $X_j$ is absent versus when it is unknown. There seems no change in magnitude of the violation comparing the inference where $X_j$ is unknown versus when it is present (colored lines have the same slope). This is confirmed by looking at the effect of confidence on these inferences, as the effect is larger when $X_j$ is unknown compared to when it is absent ($\Delta$ = 4.04, $SE$ = 1.64, $t$(1163) = 2.47, $p$ = .037, first two columns Fig 10A), but it is not different when compared to when $X_j$ is present ($\Delta$ = 0.107, $SE$ = 1.65, $t$(1163) = 0.065, $p$ = .998, second and third columns Fig 10A). So higher confidence seems to lead to a larger independence violation exclusively when comparing the $P(X_i = 1|Y = 1, X_j = 0)$ and $P(X_i = 1|Y = 1)$ inferences, and seemingly not for the other inferences (Fig 10).

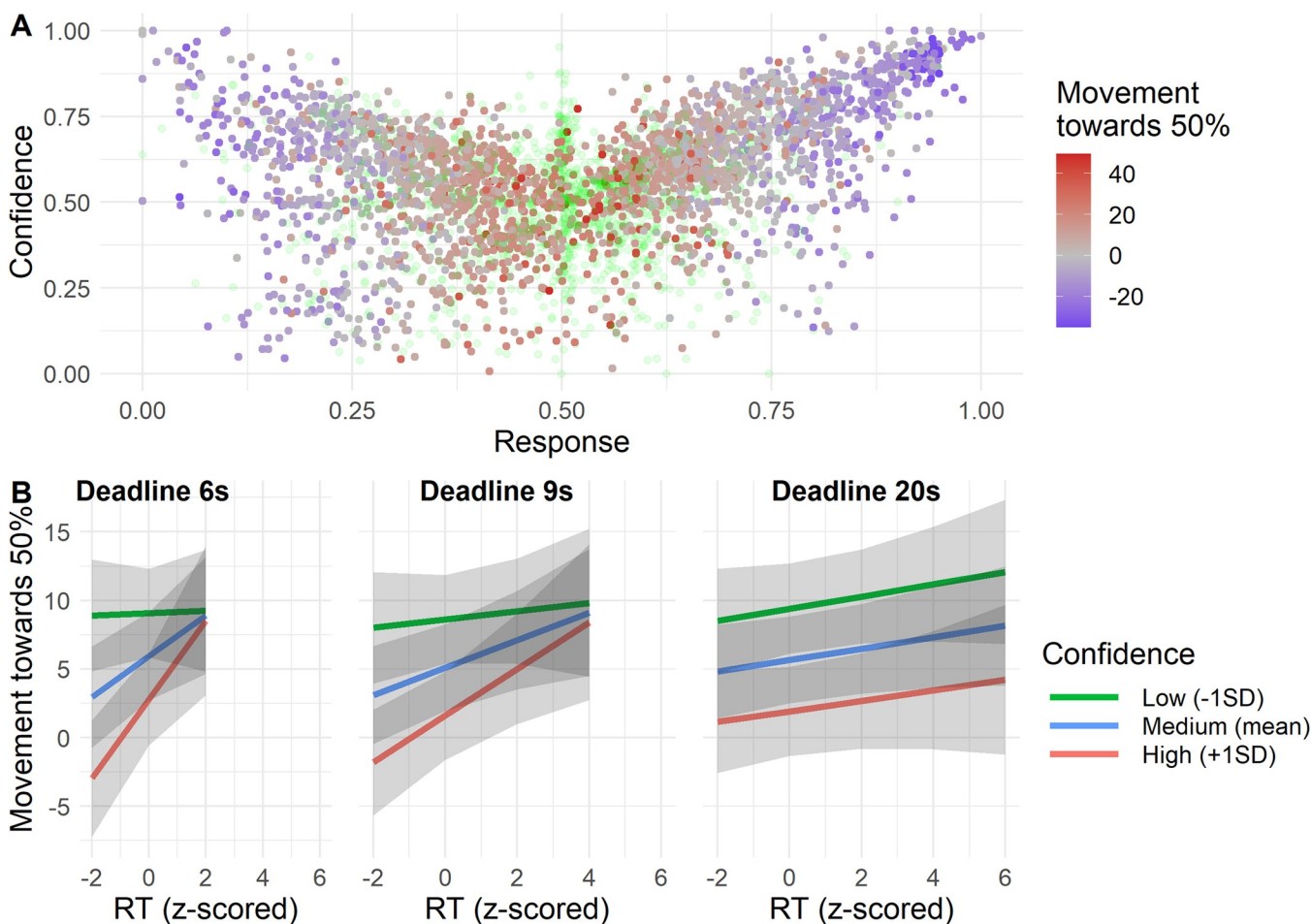

**Fig 11. Confidence and conservatism.** A. Scatterplot of responses and confidence. Responses are colored from red through grey to blue based on the distance they moved towards 50% from the normative answers (i.e. conservatism). Green transparent dots represent responses that either moved beyond 50% or for which the normative answer was 50%, and so cannot be color-coded based on their movement towards 50%. B. Plots of interaction effect of deadline, RT, and confidence on response movement towards 50%. Based on regression model discussed in text. Grey ribbons indicate 95% confidence interval.

*3.3.3.2 Common effect structure and explaining away.* We did not find an effect of confidence on Markov violations in common effect structure ($F(2, 263) = 0.123$, $p = .88$, $BF_{01} = 11.0$), nor did we find an effect on explaining away ($F(2,273) = 0.563$, $p = .57$, $BF_{01} = 15.4$). All in all, it seems that Markov violations or failures to explain away are not systematically related to the confidence participants have in their responses.

**3.3.4 Confidence and conservatism.** To analyze the role of confidence in conservative inferences, we again used the same metric of conservatism as in Experiment 1, which indexes how much a response moved from the normative answer towards 50%. Fig 11A plots participants responses against confidence, which indicates that more extreme responses were associated with higher confidence.

We conducted a regression on conservatism using RT, Deadline, confidence and all interactions as predictors. We found mixed evidence for a three-way interaction ($F(2, 1668) = 3.16$, $p = .043$, $BF_{10} = 0.328$, Fig 11B). From Fig 11B it seems that participants responded more conservatively the closer they got to a deadline, and this effect is strongest for the shorter deadline conditions. Tentatively, this effect seems to be moderated by confidence. Responses with low confidence were generally already conservative regardless of RT and time pressure, while those

responses with high confidence moved further towards 50% the closer they got to the deadline (see Fig 11B). Most importantly, we found strong evidence for a main effect of confidence ($F$(1, 1675) = 236, $p < 0.001$, $BF_{10} > 100$, β = -3.46, $SE = 0.436$), indicating that responses with lower confidence tended to be closer to 50%. This last point is visible from the colors in Fig 11A, which indicate how much responses moved towards 50%.

### 3.4 Discussion

We replicated the main findings of Experiment 1 in an online replication experiment. We again found that Markov violations and failures to explain away are not impacted by time pressure. We did find an effect of RTs on explaining away, but this, like in Experiment 1, seems to be better explained by conservative responding. Participants decreased their response, getting closer to 50%, with longer RTs on the inferences where the non-queried cause was present and when it was unknown. Conservatism on these inferences should be more pronounced than for the inference where the non-queried cause is absent as the mean responses are more extreme for the former. Additionally, as the mean responses to the inference were below the normative response (Fig 9C), the effect of RT means that these responses are further away from the normative response when RTs are longer. If the effect of RT was due to participants more properly explaining away, we would expect this inference to increase, which is the opposite of what we find.

We also replicated the effects of time pressure on conservatism. Participants respond more conservatively when pressure to respond increases (i.e. they approach the temporal deadline). In agreement with Experiment 1, it seems that conservative responses are qualitatively different from Markov violations and failures to explain away.

Such a conclusion is corroborated by our analysis of confidence reports, which showed that low confidence is associated with conservative responding, while not being systematically related to Markov violations or failures to explain away. We did find an effect of confidence on Markov violations but only when comparing the $P(X_i = 1|Y = 1, X_j = 0)$ and $P(X_i = 1|Y = 1, )$ inferences in the common cause and chain structures (Fig 10). The difference between these inferences was larger when confidence was high. However, this effect was not present for the other inferences in the common cause and chain structures, nor for the Markov related inferences in the common effect structure. If higher confidence was truly related to larger Markov violations we would expect the effect of confidence to be present for all Markov violations, not just the aforementioned two inferences.

We did find a systematic relationship between confidence and conservatism. Low confidence was associated with increased conservatism, regardless of time pressure (see Fig 11B). Responses high in confidence, however, seemed to vary in conservatism (red lines in Fig 11B).

Lastly, it is notable that there was a substantial drop-out rate as in Experiment 1 due to the exclusion criterion on overall error. As discussed in the introduction to Experiment 2, while such dropout rates are not uncommon for demanding reasoning tasks [e.g. 16, 84–86], it was a reason for us to replicate the findings from Experiment 1. Experiment 2 had a larger dropout rate than Experiment 1, which can be expected given the possibility of lower task compliance when conducting online experiments [98–101]. A supplementary analysis of the excluded participants indicates that they did not comply with task instructions or did not understand the task (see S1 Appendix). Most importantly, we replicated the main findings from Experiment 1 using an online task. Moreover, as the participants included in the analysis have an overall error below 18% due to the a priori exclusion criterion, we know that they are performing above chance. This means that they understand the task to a certain degree and are engaged in

causal inference as intended. Thus, at minimum, our findings are stable for the subpopulation of people that understand the task and comply with task instructions.

## 4. General discussion

### 4.1 Summary of findings

Our experiments were aimed at elucidating the cognitive effect of time pressure on causal reasoning. To this extent we asked participants to draw causal inferences from known causal structures and manipulated the available time to respond while measuring RT.

We found that time pressure led to quicker and less accurate responses, i.e. we found an overall macro-SAT, in line with numerous studies on other types of reasoning and decision-making [63, 64]. We also established that overall performance decreased with response time, a micro-SAT [64]. Generally, the micro-SAT can take different forms, sometimes slower RTs are associated with being less accurate and sometimes they are associated with being more accurate [64]. In our case we find that slow responses are less accurate, indicating that the possible benefit of having more deliberation time is outweighed by an increase in time pressure due to the passing of time. This leads to the micro- and macro-SATs being in opposite directions, meaning that whether longer RTs are associated with improved accuracy depends on whether one looks within or across conditions [64]. In summary, while participants were overall less accurate when presented with shorter deadlines, within each deadline condition their least accurate responses were those that took the longest amount of time.

Additionally, we investigated the effect of time pressure on persistent patterns of non-normative responding: Markov violations, failures to explain away, and conservative inferences [2, 32, 33, 38]. We found that overall, the magnitudes of neither Markov violations nor failures to explain away are affected by time pressure, neither on the macro- nor on the micro-SAT level. This is in line with the conclusion in [18] that such violations can be the result of careful and deliberative reasoning.

In contrast, the magnitude of conservatism is impacted by time pressure. These conservative responses were more common under stronger time pressure and conservatism increased when participants approach the response deadline. This effect is stronger for shorter deadlines. As this response pattern appears similar to the overall SAT effects, it is plausible that the overall SAT is due to changes in the amount of conservatism. This is corroborated by our analysis of confidence in Experiment 2, where we find that both conservative responding and overall accuracy are related to participants' confidence in their judgments. Conservatism in responding was most severe when participants were uncertain about their judgments. Hence, we conjecture that the pattern of conservative responding we found is due to uncertainty and experienced time pressure. This would explain why participants respond more conservatively the closer they get to a deadline, and why this effect is substantially smaller in the longest deadline condition, where participants rarely responded close to the deadline.

### 4.2 Potential explanations of causal reasoning

What are the implications of these findings for existing theoretical accounts? The first thing to note is that the finding that Markov violations and failures to explain away are not sensitive to time pressure is surprising for a number of reasons. Firstly, accuracy in judgement and decision-making typically decreases under time pressure [63, 64]. Secondly and more importantly, it seems that promising explanations of exactly these violations predict that they would increase; this includes the Mutation Sampler [62], and heuristic-based explanations [18, 33, 38] including the Quantum Probability theory [102].

**4.2.1 Sampling theory.** Davis and Rehder [62, 103] proposed a theory of causal reasoning that accounts for the normative violations based on a sampling procedure. This work built upon recent developments in cognitive science that propose sampling schemes to underlie a variety of probabilistic judgements in different domains [104–107]. The model, termed Mutation Sampler, proposes that people engage in sampling over states of a causal network to make inferences. Subsequently, people compute inferences based on the relative frequencies of events in the generated samples.

The Mutation Sampler mechanism posits that Markov violations and failures to explain away are due to a biased and limited sampling procedure. Deadline-induced time pressure would further limit this sampling procedure, making the bias more pronounced, leading to larger violations. Similarly, longer RTs would indicate a longer sampling procedure, reducing bias and hence these violations. In this way the model would predict that the effect of deliberation time would overcome that of internal time pressure. However, neither of these predictions is borne out in our experiments.

Consistent with our findings, the Mutation Sampler does predict an increase in conservative responding due to time pressure. The prediction is not directly due to the bias in sampling but relates to the probability of sampling from the causal states necessary to calculate the required relative frequency. If these states are not sampled, the Mutation Sampler predicts a default response of 50%. When presented with less sampling time (e.g., due to a response deadline), the number of trials on which these critical states are not yet sampled increases. This would result in an increase of responses centered at 50%. In both our experiments we find spikes at 50%; participants responded with 50% on 5.1% of all trials. To test if the spikes change due to the deadlines, we conducted a repeated-measures ANOVA using the frequency of response between 49.5% and 50.5% as dependent variable. We find that the deadlines have no effect on the size of the spikes ($F(2,84) = 1.46$, $p = .239$, $BF_{01} = 4.03$). This suggests that the effect of time pressure on conservative responding cannot be attributed to an increase of responses at 50%. It should be noted, however, that the Mutation Sampler is not the only possible implementation of a sampling approach to causal reasoning [62]. For instance, the Mutation Sampler could be modified by incorporating a prior probability distribution that weights responses near 50% more strongly, which would be able to predict conservatism not just by spikes of responses at 50%. This idea is elaborated upon in Section 4.3.

**4.2.2 Heuristics and biases.** There exist multiple bias- and heuristic-based explanations of the normative violations we have discussed in this article, including an associative reasoning bias [18], the rich-get-richer bias [38], the monotonicity assumption, conflict aversion, ambiguity aversion [33], and the Quantum Probability model [102]. Except for the associative bias [18], the authors of these explanations have not explicitly considered predictions related to time pressure as they present their theories as descriptive (in contrast with the Mutation Sampler, which is a process model). However, typically the reliance on heuristics and biases increases when people are under time pressure [108–112]. Hence, if these proposed heuristics and biases function as typical heuristics and biases do (the authors provide no reasons for why we should not expect them to), we would expect that the increased reliance on them due to time pressure would result in larger normative violations. For example, Rehder and Waldmann conclude that people's inferences are "a product of an interaction between the normative model and the rich-get-richer principle" [38]. Assuming that the richer-get-richer bias functions as a typical bias, we would expect the relative contribution of the richer-get-richer bias to increase under time pressure. On the other hand, when there is little time pressure, participants would be able to engage in more deliberative strategies that would decrease the reliance on heuristics and biases, and thus decrease violations. Similarly, a dual-systems perspective that attributes the reasoning errors to System I responses (such as discussed in

[18]), would also wrongly predict an increase of these errors under time pressure as time pressure is known to increase intuitive responding [60, 113]. Again, these accounts, like the sampling account, predict that the effect of an increase in deliberation time would outweigh the negative effect of internal time pressure on accuracy.

While we do find an effect of time pressure on overall accuracy, we do not find a systematic effect on Markov violations and failures to explain away, which is not consistent with this perspective on the working of biases and heuristics under time pressure. We could speculate that some heuristics are more affected by time pressure than others. Heuristics and biases that explain Markov violations and failures to explain away by inducing correlations between variables (the associative bias [18]; the richer-get-richer principle [38]; the monotonicity assumption [33]) could be more resilient to time pressure. These heuristics can be implemented by the simple tallying of positive and negative cues in the stimulus, i.e. a tallying strategy [114], which could be such a fast and automatic strategy that it is not affected by time pressure. Rehder [18] discusses the idea that associative reasoning might not be affected by more extensive deliberation, especially when reasoners are not confronted with a cue that they are wrong, because associative responses are so easy to generate. In addition, he raised the possibility that while being able to reason causally, people might often lack the metacognitive awareness that associative and causal reasoning might result in different responses. A possible explanation is then that the heuristics accounting for conservatism, like ambiguity and conflict aversion [33], might be more sensitive to time pressure as they are directly related to uncertainty which is affected by time pressure. However, these heuristics also partly predict Markov violations and failures to explain away. Hence it is unclear whether they are part of the right explanation, since if these heuristics are truly responsible for conservatism and are affected by time pressure, we should have observed a systematic impact of time pressure on Markov violations and failures to explain away.

## 4.3 Different classes of violations and implications for theories of causal reasoning

None of the theories just discussed seem to be consistent with all our observations. Nevertheless, our results do point to a way forward. Our results indicate that not all systematic non-normative reasoning patterns in causal reasoning are the result of a single cognitive process or mechanism.

The sensitivity of conservative inferences to time effects and their relationship to confidence suggest that they have a different source than Markov violations and failures to explain away. This seems probable considering that these errors are of different types. While Markov violations and failures to explain away refer to the not adhering to normative (in)dependence relationships between certain causal variables, conservative inferences are not related to such (in) dependencies stipulated by CBNs. Moreover, Markov violations and failures to explain away are relational in the sense that they require the comparison of multiple judgments, while conservatism is measured only in comparison with a normative response. In light of these theoretical considerations and our results, it seems clear that we should view Markov violations and failures to explain away as belonging to a different class of errors than conservatism. This is at odds with existing theories of causal reasoning that attempt to explain all three errors partly with the same mechanism (the Mutation Sampler in [62]; Beta Inference, Conflict, and Ambiguity Aversion in [33]).

Our findings suggest that conservative inferences could be the outcome of a more general phenomenon related to uncertainty. Indeed, conservatism has been found in a wide variety of tasks in which participants have to judge probabilities [115–119]. When participants are

uncertain, as evidenced by confidence judgments, they might use 'default' or safe response options, such as the middle of the scale [35]. This ties into a larger issue concerning the interpretation of probabilities [120, 121]. Simply put, a response of '50%' can represent a strong belief of a participant that the correct answer is '50%', but such a response can also represent the lack of a strong belief, i.e. epistemic uncertainty. That we found conservatism to be related to uncertainty provides evidence for the latter interpretation: participants seem to respond near 50% because they are uncertain about the correct answer, not because they necessarily believe the correct answer to be near 50%. Following this line of reasoning, we can view participants' probability responses as an expression of second-order probabilities, i.e. the probability that a probability (judgement) is correct or 'epistemic reliability' [122]. This interpretation of responses around 50% is bolstered by recent findings in non-probabilistic causal judgements tasks. In these tasks, participants are asked to rate to what extent one factor caused another, and participants were found to use the middle of the scale when they were uncertain [93].

We conjecture that the conservatism we observe is due to participants using priors on the inference [123]. Priors on causal parameters (in particular causal strengths) have been proposed before [25, 33, 124, 125]. One problem with such an approach is that it can't explain differences in conservatism across inferences [33]. However, what we propose is that participants could include prior knowledge about good responses to a query. When people are presented with a stimulus, they integrate the evidence they gain from the stimulus with prior information. In Bayesian terms, the stimulus is the 'likelihood'. If they are unable to gather much evidence for a response from a stimulus, e.g. in the case of a conflict or ambiguous trial, the prior will dominate. Conversely, if the stimulus is clear-cut, providing consistent cues (as in e.g. $P(X_i = 1 | Y = 1, X_j = 1)$), its influence will dominate the posterior (that is, the judgment). This explanation would be in line with recent trends modelling cognition using Bayesian principles [126–129]. Such a mechanism could explain conservatism overall, and additionally the effect of time pressure on conservatism; with more time pressure less evidence can be gained from reasoning based on the stimulus and hence the effect of the prior on the judgment increases. In the case of extreme uncertainty, when there would be no to little incorporation of information from the stimulus, this might result in responding at exactly 50%. This could explain the spikes of responses at 50% in causal judgement studies, if we assume that the prior knowledge that participants incorporate puts an emphasis on 50%. Our results are in line with viewing confidence judgements as an indication of the relative contribution of a prior to people's judgements.

## 5. Conclusion

Our study for the first time shows that causal reasoning mechanisms are systematically affected by both external (deadlines) and internal (passing of time) time pressure. This revealed a complex pattern of macro- and micro-SAT, which can be used to test and inform theories of causal reasoning. It seems that conservative inferences are the result of a different cognitive mechanism than that responsible for Markov violations and failures to explain away, as the former is related to time pressure and confidence while the latter are not. This study therefore also emphasizes the need for a wider range of behavioral phenomena than just plain mean responses to be incorporated into theories and computational models of causal reasoning. Incorporating more detailed phenomena–like the (in)sensitivity to time pressure, confidence [93], but also between- [33, 62] and within-participant [35] variability–will lead to better theories. Other domains of judgment and decision-making have benefitted enormously from such a turn.

## Supporting information

**S1 Appendix. Analysis of excluded participants.**
(DOCX)

**S2 Appendix. Additional analyses Experiment 2.**
(DOCX)

## Author Contributions

**Conceptualization:** Ivar R. Kolvoort, Robert van Rooij, Katrin Schulz, Leendert van Maanen.

**Data curation:** Ivar R. Kolvoort, Elizabeth L. Fisher, Leendert van Maanen.

**Formal analysis:** Ivar R. Kolvoort, Leendert van Maanen.

**Funding acquisition:** Ivar R. Kolvoort, Robert van Rooij, Katrin Schulz, Leendert van Maanen.

**Investigation:** Ivar R. Kolvoort, Elizabeth L. Fisher, Robert van Rooij, Katrin Schulz, Leendert van Maanen.

**Methodology:** Ivar R. Kolvoort, Leendert van Maanen.

**Project administration:** Ivar R. Kolvoort, Katrin Schulz, Leendert van Maanen.

**Resources:** Ivar R. Kolvoort, Robert van Rooij, Katrin Schulz, Leendert van Maanen.

**Software:** Ivar R. Kolvoort, Elizabeth L. Fisher, Leendert van Maanen.

**Supervision:** Ivar R. Kolvoort, Robert van Rooij, Katrin Schulz, Leendert van Maanen.

**Validation:** Ivar R. Kolvoort, Elizabeth L. Fisher, Robert van Rooij, Leendert van Maanen.

**Visualization:** Ivar R. Kolvoort.

**Writing – original draft:** Ivar R. Kolvoort.

**Writing – review & editing:** Ivar R. Kolvoort, Robert van Rooij, Katrin Schulz, Leendert van Maanen.

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
