## [Decision Letter · Decision Letter 0]

7 Jun 2023

PONE-D-23-01378Probabilistic causal reasoning under time pressurePLOS ONE

Dear Dr. Kolvoort,

Thank you for submitting your manuscript to PLOS ONE. After careful consideration, we feel that it has merit but does not fully meet PLOS ONE’s publication criteria as it currently stands. Therefore, we invite you to submit a revised version of the manuscript that addresses the points raised during the review process. Specifically, Reviewer # 2 raises important concerns about the exclusion criteria, fast saccades, and interpretation of results in terms of the time to deadline vs. decision time. For the first concern, I suggest that you analyze data from the excluded subjects to test if they show any specific patterns or whether their behavior reflect poor performance due to certain factors (e.g., lack of motivation, misunderstanding the task, etc.). For the latter concerns, you could perhaps look at the data from the point of decision time to test whether longer deliberation time could have positive effect on the performance. Such analyses could provide a complementary account of the results, followed by appropriate discussion. The reviewers point out other concerns and suggestions that need to be addresses as well.

We look forward to receiving your revised manuscript.

Kind regards,

Alireza Soltani

Academic Editor

PLOS ONE

Journal Requirements:

Reviewers' comments:

Reviewer's Responses to Questions

**Comments to the Author**

1. Is the manuscript technically sound, and do the data support the conclusions?

Reviewer #1: Yes

Reviewer #2: Yes

2. Has the statistical analysis been performed appropriately and rigorously? 

Reviewer #1: Yes

Reviewer #2: Yes

3. Have the authors made all data underlying the findings in their manuscript fully available?

Reviewer #1: Yes

Reviewer #2: Yes

4. Is the manuscript presented in an intelligible fashion and written in standard English?

Reviewer #1: Yes

Reviewer #2: Yes

5. Review Comments to the Author

Reviewer #1: (The comments are also attached.)

In this manuscript, the authors have conducted two probabilistic causal inference experiments by manipulating time pressure. The authors found the accuracy-tradeoff pattern in the behavior, as expected, and more conservative responses in the presence of time pressure. However, they found that Markov violations and failures to explain away, used as two other measures of reasoning error, stayed the same across the two experiments. The authors suggest that confidence plays an important role in dissociating the effects and that the reasoning error should be attributed to multiple processes.

The experimental design, analyses, and findings are interesting. However, so many analyses reported in the manuscript can distract from the main findings. Therefore, focusing more on the main results and moving auxiliary analyses to the supplementary materials can help the reader to understand the take-home message more accessible from the Results section.

Minor comments:

1) In Experiment 1, what is the logic behind the exclusion criteria of 18%? Why this specific number?

2) Have the authors performed power analyses to find the number of participants (41 in Experiment 1)? If not, post hoc power analysis can be helpful.

3) It is highly recommended to report effect sizes wherever applicable to give a better sense of the significance of the difference alongside the p-value.

4) The motivation and approach could be clearer in lines 378-385. It also needs to be clear how the CBN predictions are calculated in line 381. In general, more clarification on how the CBN predictions are obtained will be helpful.

5) In Figures 3 and 4 (and other applicable figures), showing the data points (instead of only the bar plot) will also be helpful.

6) It needs to be made clear where the results in lines 388-391 are coming from or how they have been calculated.

7) Line 446 needs to be clarified what the independent variable is.

8) Line 507 needs to be revised (incomplete sentence in the middle of the line).

9) In Experiment 2, since there is a pronounced unbalance in the gender of the participants, it will be helpful to recruit more female participants. Furthermore, in Line 518, does it mean 20/37 participants were excluded, and the results are based on 17 participants?

Reviewer #2: The authors present an interesting set of experiments investigating the effect of time pressure on different kinds of errors in reasoning about causal scenarios. The investigation of time pressure on reasoning in general has many practical implications and the findings presented here would be a valuable contribution to the field. Although I think the manuscript should be published, I have a number of concerns and some minor suggestions.

My main concern is regarding the large proportion of participants who were excluded from the analyses. In order for this research to be useful, one would have to be able to generalise the results to the general population and the large proportion of excluded participants makes this difficult. Although this has happened in other causal reasoning experiments, the generalisability issue remains nevertheless. I must admit that I didn’t fully understand how an average precision value of 18% corresponded to participants always responding with 50%, perhaps some further clarification might be worthwhile. In any case, it would have been good to see the whole data, or at least data where a more lenient exclusion criterion was applied.

If I’m reading Fig 4 correctly, responses with very fast reaction times were most accurate. If so, wouldn’t this microSAT be the opposite to the macro one observed between conditions, and actually the opposite to a speed-accuracy trade-off altogether? Perhaps a brief discussion of this would be good even at that point in the manuscript, and I don’t think the effect shown in Fig 4b should be called an SAT effect since it is quite the opposite.

The authors also refer to conservatism (decisions closer to 50%) as a microSAT, but again, a speed accuracy trade-off implies more accurate decisions with longer reaction times, which is not what happened. Perhaps a different term than SAT should be used to describe these effects.

If the authors were interested in decisions made closer to the deadline, I wonder if it would be useful to plot the time remaining (i.e., deadline – RT) rather than z-scored RTs, which makes interpreting the effects more difficult. It would have been useful to plot the data points as well (though I understand those might be messy plots) and test for non-linear trends, as I suspect that very fast decisions and decisions closer to the deadline might both be more suboptimal than decisions with medium reaction times.

The authors frame their RT results in terms of time pressure (i.e., deadline – RT, or time left to respond), but nevertheless, the closer the participant responds to the deadline, the more time they would have spent thinking about their answer. So one could argue that these RT effects are contrary to the time pressure hypothesis if one focuses on the length of the RT rather than time remaining to respond. I am not arguing that the time remaining does not influence the participants’ judgments, I am just pointing out that one could focus on either the length of the response versus the length of the remaining time before the deadline and focusing on these two time periods generates opposite predictions. This makes generating some clear predictions from the models discussed more difficult. For example, one could turn many of the arguments around by referring to ‘rushed responses’ (short RTs) rather than internal time pressure (long RTs closer to the deadline). Although the results suggest that people may have experienced some sort of deadline pressure, had anyone made predictions from these models before seeing these results they might have made opposite predictions.

Minor comments:

I would suggest using the word ‘error’ rather than ‘precision’, since it better reflects what was measured (precision implies better performance at larger values, whereas error more accurately reflects deviance from a normative answer).

Perhaps the notation in the legend of Fig. 6 should follow the notation used in the previous figures (e.g., p(Xi|Y, Xj = 0) rather than non-queried cause absent).

The sentence ending with ‘…a probability that was in the oppositive half of the measurement scale as the normative response’ (page 20) was unclear to me, though the explanation that followed was clear.

6. PLOS authors have the option to publish the peer review history of their article (what does this mean?). If published, this will include your full peer review and any attached files.

Reviewer #1: No

Reviewer #2: No

---

## [Author Response · Author response to Decision Letter 0]

18 Sep 2023

The response to reviewers is uploaded as a document.

---

## [Decision Letter · Decision Letter 1]

7 Nov 2023

PONE-D-23-01378R1Probabilistic causal reasoning under time pressurePLOS ONE

Dear Dr. Kolvoort,

Thank you for submitting your manuscript to PLOS ONE. After careful consideration, we feel that it has merit but does not fully meet PLOS ONE’s publication criteria as it currently stands. Therefore, we invite you to submit a revised version of the manuscript that addresses the points raised during the review process. More specifically, Reviewer # 2 has a few minor concerns (see below) that need your attention and response.In addition, it would useful to create a subsection with header for "Data Availability" (under Methods) to make it easier for the readers to find the data and analysis code. Please note that your response and revised manuscript will not be sent out to review again and only will be evaluated at the editorial level. 

We look forward to receiving your revised manuscript.

Kind regards,

Alireza Soltani

Academic Editor

PLOS ONE

Journal Requirements:

Reviewers' comments:

Reviewer's Responses to Questions

**Comments to the Author**

1. If the authors have adequately addressed your comments raised in a previous round of review and you feel that this manuscript is now acceptable for publication, you may indicate that here to bypass the “Comments to the Author” section, enter your conflict of interest statement in the “Confidential to Editor” section, and submit your "Accept" recommendation.

Reviewer #1: All comments have been addressed

Reviewer #2: All comments have been addressed

2. Is the manuscript technically sound, and do the data support the conclusions?

Reviewer #1: Yes

Reviewer #2: Yes

3. Has the statistical analysis been performed appropriately and rigorously? 

Reviewer #1: Yes

Reviewer #2: Yes

4. Have the authors made all data underlying the findings in their manuscript fully available?

Reviewer #1: Yes

Reviewer #2: Yes

5. Is the manuscript presented in an intelligible fashion and written in standard English?

Reviewer #1: Yes

Reviewer #2: Yes

6. Review Comments to the Author

Reviewer #1: Thanks to the authors for addressing the comments raised by the both reviewers. Nice work on the revision.

Reviewer #2: I think the authors answered all my queries and made the relevant changes or explained why they chose not to. I only have a few minor comments.

The choice of 18% error as the cut-off to exclude participants was still difficult to understand, though I think I’m a little closer to understanding it. I presume it has something to do with the normative responses shown in Figure 5, if so, I think it should be clearer to the reader, one might even refer to the figure. And if I understood this criterion correctly, I agree that errors larger than 18% might mean that participants are systematically making judgments that are inconsistent with probabilistic reasoning. The percentage of fast responses is quite high for the excluded participants, and I would also report the average reaction time or the distribution of RTs, since even RTs of 2-3 s might be quite fast for complex reasoning problems.

I wasn’t sure what the authors meant by ‘…When participants failed to answer before the deadline, which happened on 19 trials in total, …’ Do they mean it happened on 19 test trials on average across participants?

I still think SAT is a poor choice of terminology given the meaning of ‘trade-off’, but I understand that the authors chose to be consistent with previous work. The additional explanations in the results and discussion sections regarding the direction of the relationship between RT and accuracy were useful and should clear up any potential misunderstandings.

On a minor note, the scrambled figures were somewhat difficult to work with and I couldn’t find Figure 8 (though I found it in the original submission).

7. PLOS authors have the option to publish the peer review history of their article (what does this mean?). If published, this will include your full peer review and any attached files.

Reviewer #1: No

Reviewer #2: **Yes: **Irina Baetu

---

## [Author Response · Author response to Decision Letter 1]

19 Dec 2023

Please see attached document "Response to Reviewers"

---

## [Editor Report · Decision Letter 2]

27 Dec 2023

Probabilistic causal reasoning under time pressure

PONE-D-23-01378R2

Dear Dr. Kolvoort,

We’re pleased to inform you that your manuscript has been judged scientifically suitable for publication and will be formally accepted for publication once it meets all outstanding technical requirements.

Kind regards,

Alireza Soltani

Academic Editor

PLOS ONE
---

## [Editor Report · Acceptance letter]

26 Jan 2024

PONE-D-23-01378R2 

PLOS ONE

Dear Dr. Kolvoort, 

I'm pleased to inform you that your manuscript has been deemed suitable for publication in PLOS ONE. Congratulations! Your manuscript is now being handed over to our production team.

Kind regards, 

on behalf of

Dr. Alireza Soltani 

Academic Editor

PLOS ONE